# Learning State Reachability as a Graph in Translation Invariant Goal-based Reinforcement Learning Tasks

**Hedwin Bonnavaud**                                    *hedwin.bonnavaud@isae-supaero.fr*
*ISAE-SUPAERO & ONERA/DTIS, Université de Toulouse, France*

**Alexandre Albore**                                    *alexandre.albore@onera.fr*
*ONERA/DTIS, Université de Toulouse, France*

**Emmanuel Rachelson**                                    *emmanuel.rachelson@isae-supaero.fr*
*ISAE-SUPAERO, Université de Toulouse, France*

**Reviewed on OpenReview:**

## Abstract

Deep Reinforcement Learning proved efficient at learning universal control policies when the goal state is close enough to the starting state, or when the value function features few discontinuities. But reaching goals that require long action sequences in complex environments remains difficult. Drawing inspiration from the cognitive process which reuses learned atomic skills in a global planning procedure, we propose an algorithm which encodes reachability between abstract goals as a graph, and produces plans in this goal space. Transitions between goals rely on the exploitation of a learned policy which enjoys a property we call *translation invariant local optimality*, which encodes the intuition that goal-reaching skills can be reused throughout the state space. Overall, our contribution permits solving large and difficult navigation tasks, outperforming related methods from the literature.

## 1 Introduction

Model-free Reinforcement Learning (RL) has demonstrated an outstanding ability to learn complex optimal policies from raw interaction data, for well-defined atomic tasks involving relatively short time and state space outreach, such as balancing a pendulum (Barto et al., 1983), learning to walk for a quadruped (Kimura et al., 2002), or learning to balance a bicycle (Randløv & Alstrøm, 1998).

But when it comes to solving more structured, long-term tasks, such as navigating through a building or a maze, baking a cake, or assembling furniture, hierarchical methods that separate global planning and local learned controllers may appear more appropriate (Eysenbach et al., 2019; Levy et al., 2019; Ichter et al., 2020). It is notable that often (although not always), atomic tasks enjoy a property which we call *translational invariance*, ie. learned policies can be re-used in unexplored states, without further computation, to reach local goals. Balancing a bicycle, for instance, implies in practice an optimal policy that recommends the same sequences of actions regardless of the geographical position, mostly because gravity does not change too much across the globe and that we ride bicycles on surfaces that have close enough friction properties. Similarly, when navigating in a reasonably homogeneous environment, reaching position $B$ from position $A$, can be achieved by applying the same policy than reaching position $B + \Delta$ from position $A + \Delta$, provided there are no obstacles in the way. The optimal policies might somehow differ, but are close enough in many practical cases. In this paper, we consider environments which enjoy this translational invariance property for local, atomic goal-reaching tasks. In such tasks, this invariance property enables pre-training action primitives under the form of local goal-reaching policies, and then transferring and composing them to solve

---

Code: https://github.com/SuReLI/TopologyLearning

complex tasks. The typical example of such environments is navigation tasks, upon which we focus in this work. Nonetheless, the framework we propose is more abstract and is designed to generalize to all cases of translational invariance. We exploit this property to efficiently learn an abstract model that is used by the agent to plan its course of action. Our contribution is threefold.

- We propose a generic framework linking goal spaces and state spaces for goal-reaching policy optimization.

- We formalize the notion of re-usability of a goal-reaching policy throughout the state space as one of translation invariance.

- We propose a complete graph-based model learning method, which relies on planning in the goal space, and chains local application of translation invariant goal-reaching policies. By combining planning and RL, this method permits solving tasks over long horizons, a common pitfall for classical RL methods.

As such, the proposed algorithm owes to several different inspirations. First, it belongs to the family of *goal-based* RL methods. Since it couples planning and RL, it also connects with *hierarchical* RL. It also abstracts *navigation* problems into a more generic class of sequential decision problems. Finally, it presents many similarities with the *Search on the Replay Buffer* (SoRB) algorithm (Eysenbach et al., 2019) and subsequent works, with several key differences which can be seen as a generalization of SoRB and permit better applicability. Section 2 sets the necessary background and puts our contribution in perspective of the current related literature. Section 3 introduces key ingredients, namely a formal definition of goals as state abstractions, a characterization of policy translation invariance, and finally the reachability graph learning (RGL) procedure. Sections 4, 5, and 6 evaluate RGL empirically, assess the contributions of its different components, discusses its properties and its ability to scale up to large and difficult domains. Finally, in Section 7 we draw some conclusions and perspectives on the presented work.

## 2   Background and related work

**Goals in Reinforcement Learning (RL).** RL (Sutton & Barto, 2018) considers the problem of learning an optimal decision making policy for an agent interacting over multiple time steps with a dynamic environment, modeled as a Markov Decision Process (Puterman, 2014) of unknown transition and reward models. At each time step, the agent and the environment are described through a state $s \in S$. When an action $a \in A$ is performed, the system transitions to a new state $s'$, while receiving a reward $r(s, a)$. Stochastic Shortest Path problems are a particular class of MDPs which aim at reaching a terminal goal state as quickly as possible. Such problems can be encoded as MDPs featuring $-1$ rewards for all transitions but those to a terminal goal state. One can quantify the efficiency of a policy $\pi : S \to A$ in every state $s \in S$ via its value function $V^\pi(s) = \sum_{t=0}^\infty \gamma^t r(s_t, \pi(s_t))$, with $\gamma \in [0, 1)$ a discount factor on future rewards (which can also be interpreted as a stepwise probability of non-termination).[1] Training an RL agent consists of finding a policy with the highest possible value function. A long-standing goal in RL is to design multi-purpose agents, able to achieve various goals through a single goal-conditioned policy $\pi(s, g)$ (Kaelbling, 1993), where the goal $g$ is either a single state in $S$ or an abstraction for a set of states. The ability of deep neural networks to approximate complex functions has triggered a renewal of interest in learning universal value function and policy approximators (Schaul et al., 2015), $V(s, g)$ and $\pi(s, g)$ respectively. Among the many approaches developed to learn goal-based policies and value functions, Hindsight Experience Replay (Andrychowicz et al., 2017, HER) proposes a seminal method which defines goal-based reward functions by re-labelling states collected in past trajectories as goals.

**Hierarchical RL (HRL).** Combining local goal-reaching sequences of actions in order to achieve a more general goal is the core idea of HRL (Sutton et al., 1999; Precup, 2000; Konidaris & Barto, 2009). Notably, among recent works, Kulkarni et al. (2016) define a bi-level hierarchical policy, using a DQN (Mnih et al.,

---

[1]SSPs are well-defined for $\gamma = 1$ but this is not the case for all MDPs so we keep this discount factor for the sake of genericity in further developments.

2015) agent to select high-level goals, that define options which make use of a low-level goal-based DQN agent. Nachum et al. (2018) specializes this idea to the case when the lower-level policy learns to achieve goals that encode relative changes to the current state. Levy et al. (2019) couples HER with a three-level hierarchy into an architecture called Hierarchical Actor-Critic (HAC).

**Learning abstract representations for planning.** An alternative to crafting a hierarchy of learned policies is to rely on RL for producing "lower level" option policies, and on some model of how these options affect the environment. The aim is then to optimize a sequence of options, or skills, which are an abstraction of actions, in a global plan, defined for an abstraction of states (which can either be learned or can be defined by the goal space). The key to such approaches hence relies on how the model is built. Silver et al. (2017) train a "predictron" which, for a given task, predicts $n$-step returns and long term values from any state, using a network that builds a consistent internal representation (the state abstraction) of the environment's dynamics and rewards. Similarly, several approaches (Ha & Schmidhuber, 2018; Schrittwieser et al., 2020; Hafner et al., 2020; 2021) build models that emulate the dynamics and rewards related to a task, and permit planning by simulating this surrogate model, but without a hierarchy of options and for a single task. In contrast, Nasiriany et al. (2019) optimize a sequence of reachable intermediate goal states (represented in the latent space of a variational auto-encoder on states) in order to reach a final goal (single task), using a pre-computed reachability metric for a given goal-based lower-level policy. Parascandolo et al. (2020) optimize online a similar curriculum of subgoals between a starting state and a given goal. They implement a divide-and-conquer approach by building an AND/OR search tree. Each node corresponds to a new subgoal in the sequence. They explore this tree with a Monte Carlo tree search strategy, which exploits the value function of a pre-trained goal-based policy. Some methods store explicitly these links between subgoals by constructing a reachability graph (which encodes transitions between abstractions of states, using abstractions of actions). In turn, this graph can be used for higher-level goal-based planning. Savinov et al. (2018) build this graph by randomly exploring the environment, and add a node for every encountered state, which yields a very dense graph. For a given goal, a shortest path in this graph is computed. Then a sequence of landmark subgoals is extracted so that each landmark is far enough from the previous one according to a pre-trained neural network. Eysenbach et al. (2019) introduce Search on the Replay Buffer (SoRB), which supposes the availability of a replay buffer of states and defines a graph where each state in a random subset of the replay buffer is a node. Then it uses the goal-reaching policy's value function to estimate edge weights between these nodes and finds a shortest path of state waypoints to the goal. Huang et al. (2019) propose a similar approach, but improve the node sampling strategy, and exploit a policy only to reach close goals during pre-training. SGM (Emmons et al., 2020) improves SoRB's results by pruning useless nodes in the graph, and edges that cannot been traversed by the control policy. Pruning useless nodes enables a reduction in the number of graph edges and permits a faster convergence to a close-to-optimal graph (ie. representative of actual reachability with a minimal number of nodes and edges). Chaplot et al. (2020) learn a reachability graph in a robotics navigation environment. For each new location in its graph, the agent uses its camera to estimate promising exploration directions. Aubret et al. (2021) and Ruan et al. (2022) incrementally grow a graph representing reachability, where nodes are abstractions of sets of states, using a neural network as a surrogate of the similarity between states. Similarly to the work of Aubret et al. (2021), PALMER (Beker et al., 2022) learns a projection space where the distance between the embeddings of two states represents the number of actions required by an optimal policy to reach one from the other. DADS (Sharma et al., 2019) learns a set of diverse skills, and a higher-level policy to decide the sequence of skills that have to be executed to reach a distant goal. DSG (Bagaria et al., 2021) incrementally grows a graph of options, represented as a local finite set of goal-reaching skills in each node. The present contribution uses the goal space as a state abstraction, and translation-invariant pre-trained policies as action abstractions to enable planning in a graph of goals.

**Re-using policies across states / Connection with navigation tasks.** Learning policies locally to re-use them in unexplored states has also been investigated to enable hierarchical RL. Konidaris & Barto (2007) propose to perform skill chaining on skills that are dependent on an discrete set of abstract agent contexts. These skills become reusable across contexts, but the definition of the contexts are specific to the task to solve and cannot be fully assimilated to an abstraction of navigation waypoints. PRM-RL (Faust et al., 2018) and RL-RRT (Chiang et al., 2019) build a re-usable goal-conditioned policy, and use it to navigate through a graph built using respectively PRM (Kavraki et al., 1996) and RRT (LaValle, 1998),

hence providing an abstraction of navigation features into a more general framework. Note however that none of these methods provide a generic and formal definition of policy re-useablity.

**Originality of the present work.** With respect to this general body of work, our contribution has several key features. We formalize a set of necessary conditions defining a context which alleviates the need to train the lower-level goal-conditioned policy on all states and goals, and enables policy re-use. Similarly to SoRB, we exploit the policy's value function as a local reachability measure, while introducing a level of abstraction since we clearly distinguish between goals and states. Similarly to DSG, we incrementally explore the environment and build a planning graph for chaining local skills, but the pre-training, exploration and graph growth strategies are different (expanded discussion in Appendix I). As developed in the next sections, this provides a sparser, abstract planning graph, closer to a hierarchy of options. Also in contrast to SoRB and SGM, we do not rely on a pre-existing replay buffer and avoid defining nodes over an arbitrary subset of sampled states; instead we incrementally grow a reachability graph to cover the attainable goal space.

## 3 Learning a reachability graph to chain translation invariant local policies

We aim to solve large goal-based MDPs, requiring long sequences of actions, and where locally trained policies can be re-used throughout the state space. The rationale for the method we propose below goes as follows. Because they are *continuous* universal approximators, neural networks are intrinsically unsuited to approximate abrupt environment of skill changes, and discontinuous functions such as the value functions arising in some difficult RL environments (e.g. mazes, non-holonomous robots, etc.). For instance, in mazes featuring thin walls, the value function is discontinuous and approximating it with a neural network propagates values through the walls which can lead to catastrophically bad greedy policies. Because neural network optimization assumes that samples are obtained independently and identically from a *stationary* distribution, they are also unsuited to retain local information: either because the distribution (and hence the training set) is unbalanced or because of distributional shift which causes catastrophic forgetting (French, 1999; Kirkpatrick et al., 2017). The sequential nature of RL decisions makes it crucial to make good decisions in infrequently-visited states, to retain local information even when facing distributional shift, and to approximate some functions that can easily be discontinuous. Neural networks are a viable policy class for low-level policies and that it may be advantageous to add additional known structure to the problem setting (which can be learned) rather than attempting to learn a complete world model with no additional assumptions which may require high sample-complexity. Elaborating on this statement, we turn to a hierarchy of approximators, coupling planning in a graph of goal space waypoints, with local goal-reaching skills learned with deep neural networks. When we would like to reach a goal $g^* \in G$ from a state $s_0 \in S$, we link $g^*$ and $s_0$ to their closest graph nodes. Specifically, we find vertices $v^*$ and $v_0$ whose waypoints $g_{v^*}$ and $g_{v_0}$ minimize some measure of proximity $d(g^*, g_{v^*})$ and $d(\mathscr{P}(s_0), g_{v_0})$ respectively, with $\mathscr{P}(s_0)$ an abstraction of $s_0$ in the goal space. Then we find a shortest path between them in the graph, which defines an *execution curriculum* of waypoints, and the local policy is used to reach each waypoint's vicinity in sequence.

The core of our contribution lies hence in the graph expansion and pruning method, its ability to accurately represent an abstraction of the environment dynamics despite unbalanced samples and discontinuous properties, and finally its use to design goal-conditioned policies over large and complex state spaces. To present the method in a well-defined framework, we restrict the set of MDPs we consider to those enjoying a property we call *translation invariance of local optimal policies* which we discuss in section 3.2. We also discuss therein to what extent this assumption is a strong constraint and how it can be related to more general transformations than translations. Then, given such a generalist goal-reaching policy $\pi$, we grow and prune a graph $\mathcal{G}$ which encodes an abstract notion of reachability and distance over the state space (Section 3.3). The pair $(\pi, \mathcal{G})$ can then be used jointly to encode a policy that benefits from the best of both worlds and allows one to exploit planning algorithms over $\mathcal{G}$ in order to define an *execution curriculum* of waypoints for $\pi$; resulting in a global agent that can reliably learn to reach distant goals in complex environments.

### 3.1 Goals as state abstractions

In the general sense, a goal $g$ is an abstraction for a set of states. For instance, a goal for a robotic ant might be "reach this room, regardless of orientation, legs configuration, or precise final position". In this

section, for the sake of genericity, we assume that goals live in a goal space $G$, that both $S$ and $G$ are normed vector spaces, and that there exists a projection $\mathscr{P}(s) = g$ which projects states into the (lower dimensional) goal space.[2] Consequently, for any goal $g$, there exists a set of states which map to $g$. Formally, we can define $K_0 = \ker(\mathscr{P})$ as the set of states corresponding to the null goal $0_G \in G$. Conversely, there is an inverse function (possibly non-unique) that provides a prototype state for any goal $g$. Let $\bar{\mathscr{P}}$ be a mapping from goals to states such that $\mathscr{P} \circ \bar{\mathscr{P}}$ is the identity function on $G$. There are many possible such mappings if the dimension of $G$ is smaller than that of $S$. Conversely, when $\dim(G) = \dim(S)$, one can take $\bar{\mathscr{P}} = \mathscr{P}^{-1}$, although in this case it is practical to straightforwardly identify goals and states, which means $\mathscr{P}$ and $\bar{\mathscr{P}}$ are the identity function. When $S$ and $G$ differ, we assume such a $\bar{\mathscr{P}}$ mapping is provided. Then, $K_g = \{\bar{\mathscr{P}}(g) + \delta, \ \delta \in K_0\}$ is the set of states whose projection by $\mathscr{P}$ is $g$. In what follows, we retain the $\mathscr{P}$ and $\bar{\mathscr{P}}$ notations for genericity, but the reader is encouraged to discard them as the identity function in order to catch the key intuitions. Finally, when the goal and state spaces differ, we introduce the strong assumption that for a given goal $g \in G$, any $s_2 \in K_g$ is reachable for a negligible cost from any other $s_1 \in K_g$. In plain words, moving between any two states which correspond to the same goal (same state abstraction) is supposed feasible and costless. Note that this is immediately verified when $S = G$.

## 3.2 Translation invariance of local optimal policies: re-using macro-actions across all states

Intuition indicates that a four-legged robot should not have to learn to walk again when it is moved from a room of the lab to another. We formalize this notion of re-usability of learned policies as one of translational invariance.[3] We say an MDP admits translation invariant local optimal policies (TILO policies) if there exists a goal-conditioned optimal policy $\pi^*$ such that

$$\forall (s, \delta) \in S \times S, \exists \rho \in \mathbb{R}, \text{ such that } \forall g \in \mathcal{B}\left(\mathscr{P}(s), \rho\right), \pi^*(s, g) = \pi^*\left(s + \delta, g + \mathscr{P}(\delta)\right), \tag{1}$$

where $\mathcal{B}(\mathscr{P}(s), \rho)$ is a ball, centered in $\mathscr{P}(s)$ and of radius $\rho$. In plain words, such a policy guarantees that whichever close enough starting states $s$ and $s + \delta$ we consider, we can always find *local* goals $g$ and $g + \mathscr{P}(\delta)$ within a distance $\rho$ of $\mathscr{P}(s)$, for which the first action recommended by the policy will be the same. A corollary of this property is that in deterministic MDPs, all actions taken to reach $g$ from $s$ are the same as those necessary to reach $g + \mathscr{P}(\delta)$ from $s + \delta$, for goals that are close enough to $\mathscr{P}(s)$. TILO policies can be trained to reach goals from any starting state, and the TILO property enables their re-usability throughout the state space to reach local goals (which marks a notable difference with the relative goal policies introduced by Nachum et al. (2018) among others). Note that considering only translation invariance is somehow restrictive as one could wish to identify invariances to other transformations (e.g. deformations on images, rotations, etc.) or the more general case of equivariance (Van der Pol et al., 2020; Mondal et al., 2022; Wang et al., 2022) which is a challenge in itself (Konidaris & Barto, 2007; Faust et al., 2018; Chiang et al., 2019) and which we reserve for future work. Appendix E discusses such a generalization.

Arguably, MDPs that admit TILO policies do not represent the full span of MDPs. However we argue that with an appropriate choice of the metric on $G$, defining $\mathcal{B}(\mathscr{P}(s), \rho)$, this property actually applies to many common control problems where the goal space supports an *addition* operation. Moreover, one can extend the reasoning to $\epsilon$-optimal policies, hence defining $\epsilon$-TILO policies. An MDP admits $\epsilon$-TILO policies if there exists a policy that is $\epsilon$-optimal and obeys equation 1. When one takes the four-legged robot that has only been trained on the lab's concrete grounds, to some other surface, it is reasonable to assume its translated goal-conditioned policy will not be optimal anymore. However, this policy is still likely to perform better than most other policies and hence to be $\epsilon$-TILO.

The method we develop herein applies to MDPs which admit $\epsilon$-TILO policies that are pre-trained. Practically, given a starting state $s$, we directly train a translation invariant goal-conditioned policy $\pi(s, g)$. To enforce the TILO property, we actually train the policy as a function of the difference $g - s$ (or $\bar{\mathscr{P}}(g) - s$ when $G \neq S$). Training of this policy is done before directed exploration and graph learning takes place. We also define a goal-proximity quasi-metric $d^\pi(g, g') = (V_{max} - V^\pi(\bar{\mathscr{P}}(g), g'))/(V_{max} - V_{min})$, indicating how close two goals are under policy $\pi$, with $V_{max}$ and $V_{min}$ chosen so that, on the training domain, $d^\pi(g, g') \in [0, 1]$

---

[2]Section 4 will relax the vector space requirement on $S$ and $G$.

[3]Appendix D discusses a generalization to other transformations.

and $d^\pi(g,g) = 0$. This quasi-metric $d^\pi(g,g')$ can be interpreted as a measure of how long it takes for policy $\pi$ to reach $K_g$ from $K_{g'}$ on average. The goal-conditioned policy training method is any algorithm that trains a universal value function approximator; it trains $\pi(s,g)$ and $V^\pi(s,g)$ within a playground task with no obstacles. For instance, starting with a random exploration strategy, a replay buffer is filled by interaction samples in the playground environement, and the goal-conditioned policy is trained to reach goals sampled from this buffer, using HER. This yields a $\pi(s,g)$ policy that is also an $\epsilon$-TILO policy, along with the corresponding $d^\pi(g,g')$ quasi-metric. We emphasize that this policy is not required to be able to reach any possible goal from $s$, even in the playground environment (Levy et al. (2019) and Nachum et al. (2018) have illustrated how RL algorithms struggle when the goals become too distant). Instead, its performance and goal outreach is as good as the training procedure can make it, and we rely on the graph learning procedure to encode the reachability between states, based on this policy.

This idea of pre-training a policy in a playground task meets the intuition of policy transfer in life-long RL. The playground stands for the previous MDPs seen in an agent's life, and the TILO property captures the idea that translation invariance is an abstraction that enables transfer between MDPs. In turn, goal-reaching policies trained in the playground stand as generic action primitives, which are then composed hierarchically when solving a new, complex task.

### 3.3 Learning a reachability topology

---

**Algorithm 1:** Reachability graph learning (RGL)

---

**1** Input: $\pi$, $d^\pi$, $\eta_{reach}$, $\eta_{node}$, $\eta_{edge}$, $T_r$, $T_e$

**2** Initialize: $V = \emptyset$, $E = \emptyset$

**3** **repeat**

**4**      $s_0 = \texttt{env.init}()$

**5**      $g_0 = $ goal associated to closest node to $\mathscr{P}(s_0)$

**6**      **if** $d^\pi(\mathscr{P}(s_0), g_0) > \eta_{edge} \lor V = \emptyset$ **then**

**7**          $V \leftarrow V \cup \{\texttt{node}(\mathscr{P}(s_0))\}$

**8**          $g_0 = \mathscr{P}(s_0)$

**9**      $v^* = \texttt{selectExplorationNode}(V, g_0)$

**10**      $(v_i)_{i \in [0,H]} = \texttt{shortestPath}(V, E, g_0, v^*)$

**11**      $s = s_0$, $t = 0$

**12**      **for** $i \in [1, H]$ **do**

**13**          **while** $\neg reached(s, v_i) \land t \leq T_r$ **do**

**14**              $s \leftarrow \texttt{env.step}(s, \pi(s, g_{v_i}))$

**15**              $t \leftarrow t + 1$

**16**          **if** $\neg reached(s, v_i)$ **then**

**17**              $\texttt{setWeight}(E, v_{i-1}, v_i, +\infty)$

**18**              **break**

**19**      **if** $reached(s, v^*) \lor H = 0$ **then**

**20**          $\{s_t\}_{t \in [1, T_e]} \leftarrow \texttt{explore}(s, T_e)$

**21**          $(V, E) = \texttt{growGraph}\left(V, E, \{s_t\}_{t \in [1, T_e]}\right)$

**22** Function $\texttt{growGraph}(V, E, \{s_t\}_{t \in [1, T_e]})$:

**23** **for** $t \in [1, T_e]$ **do**

**24**      $\texttt{addNode} = True$, $E_{in} = E_{out} = \emptyset$,

         $w = \texttt{node}(\mathscr{P}(s_t))$

**25**      **for** $v \in V$ **do**

**26**          $l_{in} = d^\pi(g_v, g_w)$

**27**          $l_{out} = d^\pi(g_w, g_v)$

**28**          **if** $l_{in} \leq \eta_{node} \land l_{out} \leq \eta_{node}$ **then**

**29**              $\texttt{addNode} = False$; **break**

**30**          **if** $l_{in} \leq \eta_{edge}$ **then**

**31**              $E_{in} \leftarrow E_{in} \cup \{\texttt{edge}(v, w)\}$

**32**              $\texttt{setWeight}(E_{in}, v, w, l_{in})$

**33**          **if** $l_{out} \leq \eta_{edge}$ **then**

**34**              $E_{out} \leftarrow E_{out} \cup \{\texttt{edge}(w, v)\}$

**35**              $\texttt{setWeight}(E_{out}, w, v, l_{out})$

**36**      **if** $addNode = True \land E_{in} \neq \emptyset$ **then**

**37**          $V \leftarrow V \cup \{w\}$, $E \leftarrow E \cup E_{in} \cup E_{out}$

**38** **return** $V, E$

---

To ease the presentation of ideas, we present the proposed Reachability Graph Learning algorithm (RGL, Algorithm 1) in the context of deterministic MDPs, and defer the discussion of the stochastic case to the end of this section. Given a pre-trained $\epsilon$-TILO policy $\pi$, we wish to construct an oriented graph $\mathcal{G} = (V, E)$ which will represent the reachability between sub-goals, using $\pi$. Each vertex $v \in V$ of such a graph is associated with a given goal $g_v$, and directed edges $e \in E$ indicate reachability of the successor node's goal from the states corresponding to the source node's goal. In other words, if an edge exists between $v$ and $w$, then $\pi$ successfully reaches $g_w$ from states in $K_v = K_{g_v}$. The edge linking $v$ and $w$ is weighted with a traversal cost of $d^\pi(g_v, g_w)$. Knowledge of this weighted graph permits running a planning algorithm to

find an execution curriculum of waypoint vertices (intermediate goals $g_v$) which eventually link any start state and final goal. This is very similar in spirit to SoRB (although our graph is defined on goals, not states). The (other) key difference lies in the fact that graph nodes are not built on an arbitrary set of sampled states, which might be rather sensitive to the distribution of these sampled states, and graph edges do not rely solely on evaluating the policy's value function, which might poorly account for discontinuities (walls) or rarely visited states. Instead we grow and prune the graph dynamically so that it actually encodes reachability between goals, which is a major difference with most other methods in the literature.

**Path planning in $G$.** During an iteration of the RGL procedure, a starting state $s_0$ is first sampled from an initial state distribution. Note that RGL does not suppose a fixed starting state. If $s_0$ is the first sampled starting state ever, or if the closest goal to $\mathscr{P}(s_0)$ lies far from $\mathscr{P}(s_0)$ in the goal space, this means $s_0$ does not correspond to any previously explored goal and we add a node in the graph at $\mathscr{P}(s_0)$. Then, a node $v^*$ in the graph is selected for exploration. This selection relies on a count-based criterion which influences the progressive coverage of the goal space. Although one could design heuristics or refined exploration strategies (Bellemare et al., 2016; Ecoffet et al., 2019; Badia et al., 2019; Burda et al., 2019; Domingues et al., 2021) for this criterion, we rely on a simple count of the number of times a node has been selected for exploration, hence promoting uniform visits to every node in the graph. A finite horizon plan $(v_i)_{i \in [0,H]}$ is computed by finding a shortest path in the graph from the starting state's node $v_0$ to the selected node $v^* = v_H$. Note that there may not exist a path between $v_0$ and $v^*$ in the graph, in which case $H = 0$ and the `shortestPath` procedure returns the single node $\{v_0\}$. Let $(g_i)_{i \in [0,H]}$ denote the corresponding sequence of waypoint subgoals. Then $\pi$ is used to sequentially reach each goal. Specifically, when trying to reach $g_v$, $\pi(\cdot, g_v)$ is run until a `reached`$(s,v) := [d^\pi(\mathscr{P}(s), g_v) \leq \eta_{reach}]$ condition becomes true, or a maximum number of steps $T_r$ is exceeded. If applying $\pi$ allowed the agent to reach the $\eta_{reach}$-neighborhood of $g_v$, then the next waypoint $w$ in the plan is selected and the procedure is repeated until the node $v^*$ is reached.

**Graph pruning.** We interpret not reaching the neighborhood of $g_w$ when applying $\pi$ as a mismatch between the notion of reachability encoded in the graph and the actual reachability in the environment using $\pi$. As a consequence, we set the cost of edge $e$ between $v$ and $w$ to $+\infty$ to account for this non-reachability. Consequently, if the graph is learned without errors, the existence of an edge $e$ between two nodes $v$ and $w$ indicates that $\pi$ permits reaching the $\eta_{reach}$-neighborhood of $g_w$ from states $s$ whose $\mathscr{P}(s)$ are in the $\eta_{reach}$-neighborhood of $g_v$ in less than $T_r$ time steps (or that this edge is never selected by the shortest path planning procedure). This pruning procedure keeps the graph free of mis-identified edges. In mazes, it deletes edges that cross walls, and hence accounts for the discontinuities we wished to represent within the policy.

**Graph expansion.** Conversely, if applying $\pi$ throughout the sequence of waypoints actually fulfills the goal $g^*$ of node $v^*$ selected for exploration, then a generic exploration procedure is performed from the reached state $s^*$ in the $\eta_{reach}$-neighborhood of $K_{v^*}$, during $T_e$ time steps. The intention of such an exploration procedure is to discover states $s$ whose $\mathscr{P}(s)$ permit expansion of the graph. We randomly sample a goal within a certain radius of $g^*$ and try to reach it using $\pi$. If we succeed, we sample another random goal and repeat this exploration until we obtain a complete exploration trajectory of length $T_e$. Note that this exploration strategy could be replaced by any other, which is why we refer to it generically as the `explore` procedure in Algorithm 1. The states visited along the trajectory are collected in a buffer. We wish to expand the graph so that its nodes induce a good coverage of the buffer states' goals and its edges indicate proximity (but not necessarily reachability at this stage). To that end, we cycle through the buffer and incrementally add vertices to $V$ whenever a goal is $d^\pi$-further away from all nodes than a threshold $\eta_{node}$. Edges are created from this new vertex to all nodes within $\eta_{edge} > \eta_{node}$. We differentiate between incoming and outgoing edges from the new candidate node: if there is no incoming edge, then the node is not added to the graph. This greedy procedure expands the graph to create new nodes that complete the goal space coverage wherever required, with limited connectivity between nodes. At this stage, some newly created edges might not account for reachability, e.g. in a maze, this might happen if the closest existing graph node to the newly created node is behind a wall. We rely on future iterations to prune the graph as presented in the previous paragraph.

Overall, this growth and pruning RGL procedure results in a graph $\mathcal{G} = (V, E)$ which encodes goal space reachability when using $\pi$ in the state space. The pair $(\mathcal{G}, \pi)$ implicitly defines a general goal-reaching policy which requires computing a shortest path in $\mathcal{G}$ to chain local executions of $\pi$ between subgoals.

**Time complexities during learning.** At execution time, determining the action to undertake in $s$ in order to reach $g$ requires solving a shortest path problem in $\mathcal{G}$. This can be typically implemented using Dijkstra's algorithm (Dijkstra, 1959), which has complexity $\mathcal{O}(E + V \log V)$.[4] It is important to note however that in deterministic MDPs (or MDPs with limited noise) this shortest path needs only be computed once per goal-reaching task and can be carried over to the next time step of the task, thus strongly dampening the overall computational cost. During learning, the pruning phase of an iteration of RGL has complexity $\mathcal{O}(E + V(\log V + T_r))$. The exploratory collection of new samples runs in $\mathcal{O}(T_e)$, while the `growGraph` function has complexity $\mathcal{O}(T_e V)$.[5] This results in an overall time complexity of $\mathcal{O}(E + V(\log V + T_r + T_e))$ for each iteration of RGL, which involves $\mathcal{O}(V T_r + T_e)$ iteration steps with the environment.

In the general case of MDPs with stochastic transitions, the pruning procedure of RGL needs to be adapted to account for the stochastic outcomes when trying to reach $g'$ from $K_g$. Note that, in this case, $d^\pi(g, g')$ captures a broader notion than the number of required time steps for $\pi$ to reach $g'$ from $g$: it captures the overall probability to reach $g'$ from $K_g$, given the transition model and a probability of termination of $1 - \gamma$ at each time step. Thus, reachability can be redefined as the probability of reaching $g'$ from $K_g$ being actually equal (or close to) $d^\pi(g, g')$. Verifying this with high confidence requires running several trials between $K_g$ and $g'$, which can be implemented by enhancing the algorithm with a memory of trial outcomes for each edge in the graph. Introducing such a delay in updates is similar in spirit to the practice of RMAX (Brafman & Tennenholtz, 2002) or Delayed Q-learning (Strehl et al., 2006), which introduce an $N_{known}$ number of samples which are necessary to correctly identify a transition. Note that in most practical implementations of such algorithms, $N_{known}$ is arbitrarily set to a small value to preserve computational efficiency. An alternative, which we do not explore here and reserve for future work, is to let the weight of an edge adapt to the observed trial durations between $g$ and $g'$.

## 4 Experimental setup

To highlight the behavior of RGL, and provide a fair and interpretable benchmark against comparable methods, we consider a set of navigation tasks in mazes, including a high-dimensional state space one.

**Environments.** In each maze, an agent should be able to reach *any goal position* from its starting point. Hence the goal space $G$ is always the 2-dimensional set of positions in the maze. We consider mazes of different complexities, with various map sizes and heterogeneous corridor widths, as illustrated in Figure 1. Namely, "four-rooms" is a $41 \times 41$-size maze resembling the classical "four-rooms" benchmark, "medium" is a more challenging maze of the same size, "hard" is an even more challenging $57 \times 57$-size maze and "mixed" has the same size as "hard" and mixes corridors and rooms of different sizes. Note that compared to mazes used in the literature (e.g. those of Eysenbach et al. (2019)), here the walls are thin, inducing sharper discontinuities in the value function across a wall. For each map, we consider three different dynamics and state spaces for the navigating agent, which we refer to as grid-maze, point-maze and ant-maze (Table 1). Grid-maze features a discrete $\{N, S, E, W\}$ action space and deterministic transitions which perform unit-length moves, hence emulating navigation on a grid. Point-maze emulates a point mass moving freely in the maze. It has a continuous, two-dimensional action space of position increments in $[-1, 1]$ on the $x$ and $y$ axes. The transitions are stochastic due to an added Gaussian noise $\mathcal{N}(0, 1)$. Contrarily to grid-maze which has a fixed starting state, point-maze randomly draws the starting point at every episode. In grid-maze and point-maze environments, the state space is simply described by the geographical position of the agent, as in the benchmarks of SoRB (Eysenbach et al., 2019) or SGM (Emmons et al., 2020), and $S = G$. Ant-maze environments build upon the MuJoCo Ant simulator (Todorov et al., 2012) and sets the ant in one of the navigation maps. Actions belong to the standard 8-dimensional action space of the Ant simulator, and the state space is the 29-dimensional space whose first two coordinates are the position of the ant's torso, as in the benchmarks of HAC (Levy et al., 2019) or Distop (Aubret et al., 2021). The transitions follow the

---

[4]For the sake of simplicity we adopt the notation $\mathcal{O}(V)$ in place of $\mathcal{O}(|V|)$.

[5]This can be amortized to $\mathcal{O}(T_e \log V)$ with efficient data structures for storing $V$ and finding nearest neighbors.

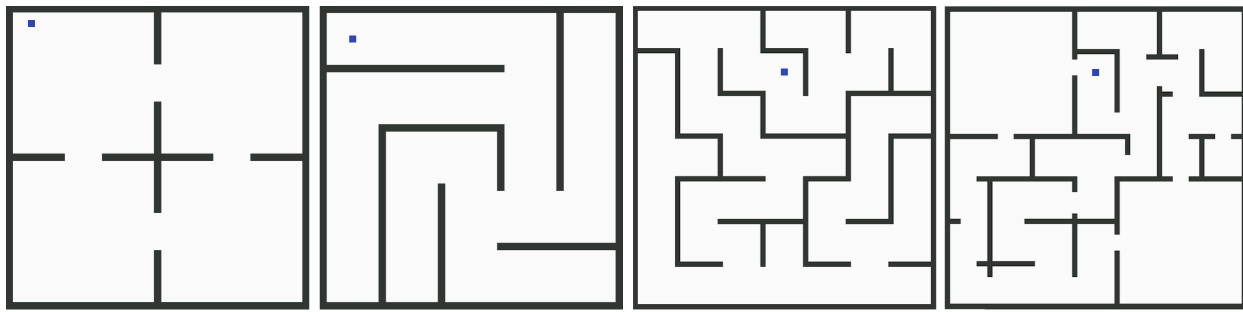

Figure 1: Mazes and starting points. From left to right: "four-rooms", "medium" (size $41 \times 41$), "hard", "mixed" (size $57 \times 57$).

dynamics of the Ant simulator. In ant-maze environments, the goal space $G$ describes only the torso's $x$ and $y$ coordinates. In all environments, agents receive a $-1$ reward at each time step, unless they reach the goal which terminates the episode. In all evaluations, every agent is independently trained 10 times. For the sake of reproducibility, the hyperparameters for all algorithms are summarized in Appendix A, and our code and results are available at [anonymous URL].

Table 1: Summary of environments

|  | $dim(S)$ | $dim(G)$ | actions | dynamics | starting state |
|---|---|---|---|---|---|
| grid-maze | 2 | 2 | discrete (number = 4) | deterministic | fixed single state |
| point-maze | 2 | 2 | continuous ($dim(A) = 2$) | stochastic | uniform distribution on $S$ |
| ant-maze | 29 | 2 | continuous ($dim(A) = 8$) | deterministic | fixed single state |

**Baselines.** To illustrate the behavior of RGL, we compare against a plain DQN agent (Mnih et al., 2015) with HER in grid-maze environments, and SAC (Haarnoja et al., 2018) with HER in point-maze and ant-maze environments. As illustrated by previous works (Nachum et al., 2018; Levy et al., 2019), such a combination can efficiently learn a goal-reaching policy for goals lying a few actions away from the starting state, but struggles to reach goals that require turning around walls. These DQN+HER and SAC+HER agents provide a baseline for performance. Another baseline consists in passing the policy learned by these base agents, along with their final replay buffer, to SoRB, to extend their outreach throughout the goal space via planning in a random subset of size $N_{init}$ of the replay buffer. Since SGM is more efficient than SoRB (due to their pruning method), we directly compare with SGM.[6]

**Ablations.** We also implement three variants of RGL. The first one is called TC-RGL, inspired by the STC method (Ruan et al., 2022), where we replace the $d^\pi$ pseudo-metric by a so-called temporal correlation network, which is an additional network trained to measure reachability between states, based on their temporal proximity during training trajectories. This variant permits evaluating the core feature of STC as an alternative to using the value function as a reachability metric between goals. The second one is called *Prune-only RGL* (PO-RGL). This version samples $n$ states from $S$ using an oracle, and sets them as graph nodes. Edges are computed in the same way than RGL. This algorithm does not need graph extension, and simply prunes the graph as it successively tries to reach random goals. Finally, NoPG-RGL (no playground) is an RGL agent whose lower-level policy has not been pre-trained in an obstacle-free playground but in the true maze, to illustrate the influence of playground environments on the pre-trained policy's quality.

**Pretraining.** First of all, it is important to note that only RGL can, by design, exploit a goal-reaching policy which has not been trained on the actual full maze (further discussion in Appendix J). This enables training RGL's goal-reaching policy in a $40 \times 40$ playground environment with no walls which facilitates pre-training (training in the full maze, as done with NoPG-RGL, is still possible but totally unnecessary since RGL's goal-reaching policy needs only navigate to local goals). Such a pre-training is not compatible with SGM or

---

[6]Note that SoRB and SGM were introduced with tailor-made, goal-based, distributional DQN and DDPG agents. We found this was not necessary for finite-length trajectories and we retain the names even though we slightly change the base agents.

SoRB. To ensure SGM builds on a sufficiently good pre-trained policy, and following the practice of SoRB, we let the base agent learn a goal-reaching policy over 300 (resp. 500) episodes in grid-maze (resp. point-maze). RGL's lower level goal-reaching TILO policy is only trained for 100 (resp. 70) episodes in the playgrounds for grid-maze (resp. point-maze) environments. Because training the temporal consistency network of TC-RGL required more samples, it was trained for 200 (resp. 600) episodes in the playgrounds. Finally, NoPG-RGL is trained for 150 episodes in the true mazes in both grid-maze and point-maze environments. To account for pre-training durations, all figures below (e.g. Figure 3) report them using vertical lines and count them within the total training time. Because ant-maze environments required specific pre-training, we defer their discussion to the end of this section.

## 5 RGL in action

This section illustrates how RGL dynamically grows and prunes its graph, illustrates the comparative importance of each part of the algorithm, and reports on its ability to reach far-standing goals.

**Visualizing graph growth and pruning.** We start by assessing separately the influence of the growth and pruning procedures on the properties of the final reachability graph. To isolate the effect of pruning, we artificially generate waypoints by using a generative model to draw states from the full state space, which yields a graph with the same number of nodes $N_{init}$ as the SGM agents (edges weights are also initialized with $d^\pi$), but with better state space coverage since the drawn states are not constrained by the exploration of the pre-trained DQN+HER agent. This graph is illustrated in Figures 2a and 2e for the "four-rooms" and "hard" grid-mazes (full results in Appendix B). In Figure 2, red edges are those whose weights have been set to $+\infty$ by the pruning procedure. We observe that (as anticipated in Section 3) only erroneous edges which were selected in a shortest path are pruned, and some remain in the graph, especially in grid-maze, which features a fixed unique starting state. This bears little consequences in terms of goal reachability since these are never selected in shortest paths from the initial state, but still result in a rather dense reachability graph. Conversely, Figures 2b-2d and 2f-2h illustrate the graph growth and the pruning for RGL in the same environment after respectively 1,000, 40,000 and 270,000 interaction steps. To avoid misinterpretations, it is important to note that since the graph is oriented, each green edge in these figures actually stands for two edges in the graph. If only one has been pruned and rendering of the other happens afterwards, the segment appears green while only one edge in the graph has non-infinite weight. Overall, the incremental growth of RGL's graph yields an efficient coverage of the state space, avoiding the clusters of unnecessary nodes of Figures 2a and 2e, and reducing the need for pruning.

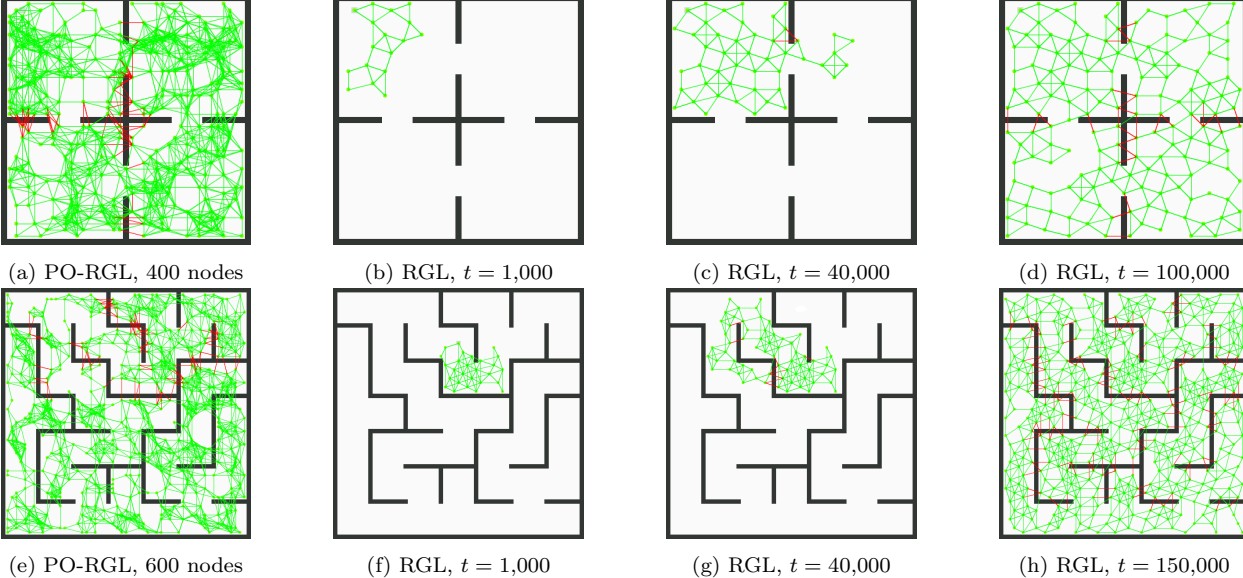

| (a) PO-RGL, 400 nodes | (b) RGL, $t = 1,000$ | (c) RGL, $t = 40,000$ | (d) RGL, $t = 100,000$ |

| (e) PO-RGL, 600 nodes | (f) RGL, $t = 1,000$ | (g) RGL, $t = 40,000$ | (h) RGL, $t = 150,000$ |

Figure 2: Reachability graphs.

**RGL agents can reach any goal.** Figure 3 reports the ability of each agent to reach any goal in the maze, along training. The vertical lines correspond to the time at which each agent starts exploiting the pre-trained policy. Every 1,000 interactions with the environment, we randomly draw 30 goals across the full goal space, and report the fraction of these goals the agent managed to reach. We call this metric the agent's *accuracy.* As expected, since exploration in mazes is difficult, the pre-training replay buffers do not cover the full state space and the baselines fail to reach all goals. Interestingly, despite the low performance of the pre-trained DQN+HER and SAC+HER agents, RGL is still able to leverage their ability to reach local goals and manages to quickly grow a goal graph which eventually covers the full maze. NoPG-RGL, despite its pre-training in the actual maze, still manages to grow a relevant planning graph and outperform SGM. It is interesting to note that this pre-training still costs less than that of SGM in number of interactions. It also yields a local policy which is strongly biased by the obstacles around the starting states. The pre-training might have learned to avoid a specific wall close to the starting point, which hinders the ability of NoPG-RGL to reach waypoints and slows graph expansion. PO-RGL displays a clear jumpstart effect in the "four-rooms" maze since its initial graph requires little pruning and most goals are readily reachable. Conversely, early planning graphs of RGL and TC-RGL contain few nodes and require expansion (and thus interaction steps) before their accuracy reaches 1. After 1,000 interactions, even though the planning graph of RGL contains only a few nodes (Figure 2b), it already reaches more goals than the baseline agents. As the mazes become more difficult, many more edges need to be pruned from PO-RGL's initial graph before it effectively represents graph reachability and the plans reliably lead to goals. This need for extended pruning is completely compensated by the sparse growth of the graph of RGL and TC-RGL; thus PO-RGL presents no advantage in terms of learning curve. In the most difficult "hard" and "mixed" mazes, the set of $N_{init}$ initial nodes of PO-RGL is just insufficient to properly cover the full goal space with feasible edges and PO-RGL's accuracy is capped around 0.5 and 0.8, while the dynamic growth of RGL permits reaching close to 1 accuracies. Also, the extra temporal consistency network of TC-RGL seems detrimental to the training process compared to RGL. Since this network only approximates the notion of reachability instead of directly using the value function, it induces a graph expansion and pruning phase with more errors or missed nodes (which were actually reachable). In turn, as TC-RGL's graph does not accurately represent reachability, some goals are eventually missed. In all environments, RGL reaches an accuracy of 1 and dominates over all variants.

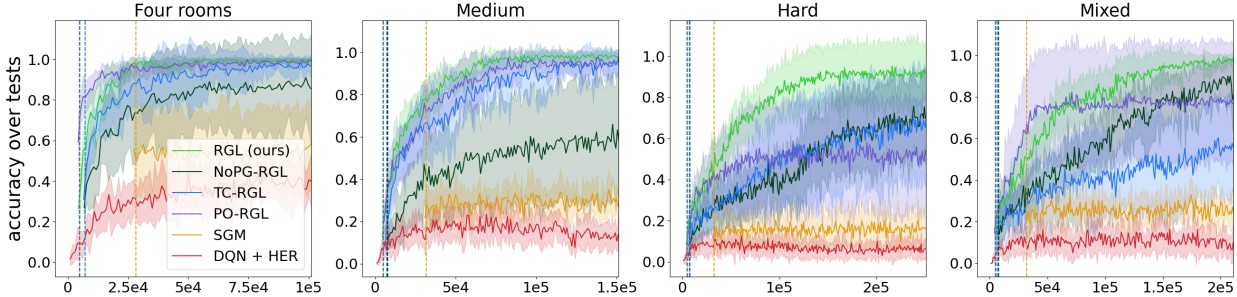

Figure 3: Accuracy for all agents on grid-maze environments, versus interaction steps. Take-aways: 1) RGL (all versions) consistently outperform the baselines, 2) the TC metric (TC-RGL) is detrimental to performance compared to the plain $d^\pi$ metric, 3) pre-training in the real maze (NoPG-RGL) can induce suboptimal local reaching behaviors which do not prevent overall coverage but limit performance, 4) online graph expansion is necessary (PO-RGL) as pre-sampled graph nodes do not match the environment's topology.

**Graph size.** Figures 4 and 5 report the number of nodes and edges for RGL agents as their graph grows in the grid-maze environments. Recall that instead of deleting edges that need to be pruned, their traversal weights are set to $+\infty$ (to avoid re-creating them again later). This is why the number of edges of PO-RGL does not decrease. Dotted curves in Figure 5 indicate the number of edges with a non-infinite weight. Overall, RGL and TC-RGL create just enough nodes to accurately represent the reachability graph given their underlying $d^\pi$ and temporal consistency networks. The relative number of nodes and edges between RGL and TC-RGL cannot be directly compared as the former uses $d^\pi$ as a distance metric while the latter uses a reachability representation, on a different (uncontrolled) scale. Still, the number of nodes is similar

|            | Four rooms   | Medium        | Hard          | Mixed         |
|------------|--------------|---------------|---------------|---------------|
| RGL (ours) | 0.99(0.02)   | **0.98(0.03)** | **0.93(0.13)** | **0.97(0.03)** |
| NoPG-RGL   | 0.86(0.26)   | 0.6(0.29)     | 0.7(0.22)     | 0.8(0.14)     |
| TC-RGL     | 0.97(0.05)   | 0.97(0.03)    | 0.68(0.26)    | 0.56(0.22)    |
| PO-RGL     | **0.99(0.01)** | 0.96(0.05)   | 0.52(0.26)    | 0.77(0.27)    |
| SGM        | 0.59(0.23)   | 0.28(0.07)    | 0.16(0.09)    | 0.25(0.09)    |
| DQN + HER  | 0.4(0.13)    | 0.14(0.08)    | 0.06(0.05)    | 0.1(0.08)     |

Table 2: Final average score for each method (standard deviation in parenthesis). Take-away: RGL statistically outperforms non-dynamic graph methods.

across mazes. Interestingly, RGL produces graphs with less connectivity, which can be interpreted as a better ability to create meaningful connections between goal waypoints for navigation, and hence sparser abstract models for planning while retaining the useful information for plan efficiency. Additionally, TC-RGL features a large variance in the number of nodes and edges developed in the graph. This appears to stem from the training of the temporal consistency network which is very sensitive to the distribution of trajectories during pre-training. In turn, this strongly affects the estimation of reachability when learning the graph and induces this variance in graph density. Appendix F provides further discussion on the impact of the graph density's hyperparameters ($\eta_{node}$ and $\eta_{edge}$) on RGL's behavior.

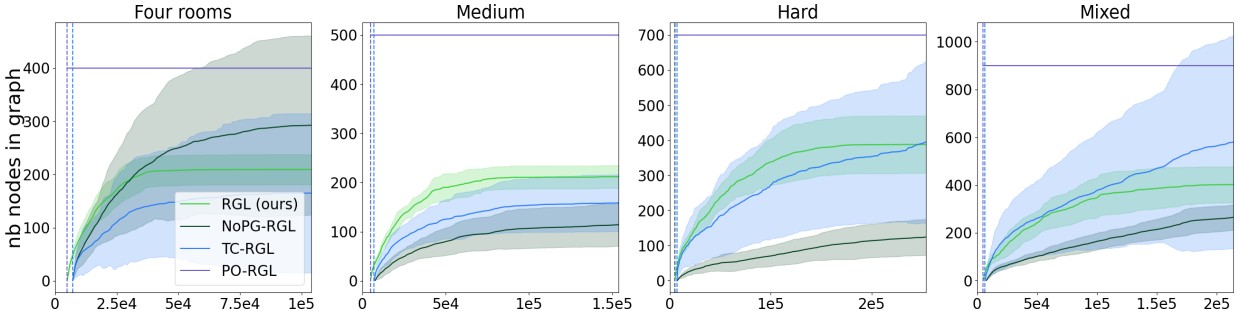

Figure 4: Number of graph nodes in grid-maze versus interaction steps. Shaded area is the $1\sigma$ confidence interval.

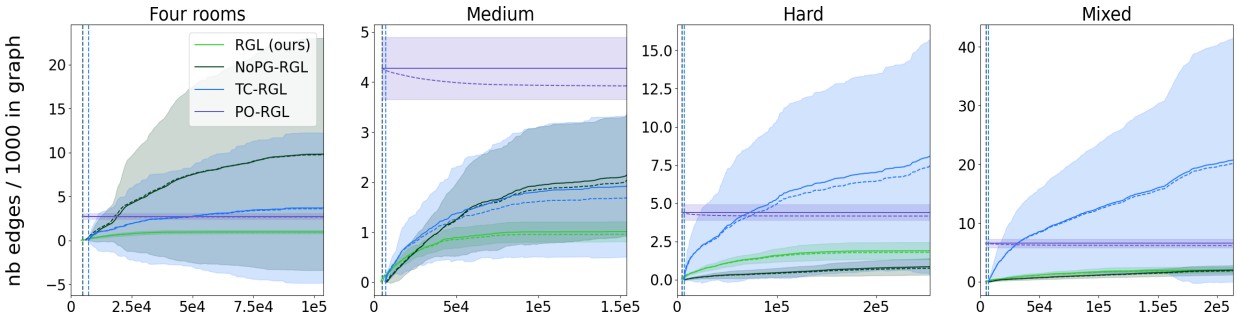

Figure 5: Number of graph edges in grid-maze versus interaction steps. Shaded area is the $\sigma$ confidence interval.

## 6 Scaling up RGL to more difficult environments

The previous section demonstrated the overall behavior of RGL and its ability to build a more relevant planning graph than similar methods. In this section, we consider additional challenges and evaluate RGL in other contexts. Namely, we explore how RGL compares to other approaches when the environment is not constrained to start in the same states every time. We also evaluate how RGL can cope with stochastic transition dynamics. Finally, we scale up RGL to the large state space of MuJoCo's ant simulator and

demonstrate its behavior when the state space is much larger than the goal space, while comparing its behavior to state-of-the-art competitors HAC (Levy et al., 2019) and DSG (Bagaria et al., 2021).

**Easier exploration in environments with the ability to "reset anywhere".** Point-maze environments permit resetting the environment anywhere in the state space at the beginning of each episode, as in the benchmarks of Eysenbach et al. (2019); Emmons et al. (2020). This feature induces diversity in the replay buffers by triggering easier exploration of the state space, simply by enabling these random resets, and somehow departs from the more constrained RL framework with a fixed (or a limited set of) starting state(s). Consequently, these environments are more favourable to SGM since their replay buffer covers a larger portion of the state space, and SGM performs better in these environments than in grid-maze ones. Figure 6 compares the behavior of all methods in the point-maze environments and illustrates how, even in this case, the graph growth of RGL eventually outperforms competing methods, as it dynamically and progressively discovers new goal waypoints to better map the state space.

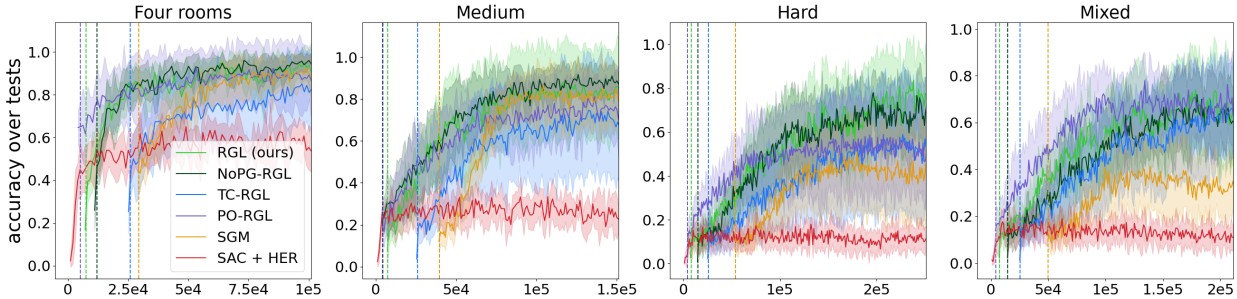

Figure 6: Accuracy for all agents on point-maze environments, versus interaction steps. Take-away: the ability to reset anywhere during training strongly helps SGM's coverage by design, but the dynamic graph growth of RGL and its variants still enables more efficient overall goal-reaching behaviors.

|            | Four rooms   | Medium       | Hard         | Mixed        |
|------------|--------------|--------------|--------------|--------------|
| RGL (ours) | 0.91(0.1)    | 0.85(0.24)   | **0.77(0.2)**| **0.67(0.23)**|
| NoPG-RGL   | **0.94(0.05)**| **0.87(0.05)**| 0.7(0.15)   | 0.61(0.15)   |
| TC-RGL     | 0.82(0.17)   | 0.68(0.27)   | 0.54(0.31)   | 0.64(0.19)   |
| PO-RGL     | 0.87(0.18)   | 0.7(0.18)    | 0.48(0.2)    | 0.64(0.17)   |
| SGM        | 0.88(0.09)   | 0.79(0.13)   | 0.43(0.24)   | 0.34(0.13)   |
| SAC + HER  | 0.54(0.1)    | 0.23(0.06)   | 0.11(0.05)   | 0.11(0.05)   |

Table 3: Final average score for each method (standard deviation in parenthesis). Take-away: RGL statistically outperforms non-dynamic graph methods, and methods that are unable to expand their graph after pre-training.

**Accommodating stochastic transition models.** As mentioned in Section 3.3, RGL in its presented version is designed for deterministic dynamics and would require some adaptations to account for transition uncertainty. Point-maze environments feature a rather high level of action noise ($\sigma = 1$ for action values in $[-1, 1]$). This makes the pruning procedure stochastic, as it will prune out edges depending of a single trial's success. Despite this rather naive behavior, RGL still manages to find paths (possibly sub-optimal) to goals and reaches a high level of accuracy (Figure 6) demonstrating a reasonable level of robustness to transition stochasticity.

**Scaling-up to high-dimensional and large state spaces, where $G \neq S$: ant-maze tasks.** A key achievement of deep RL is its ability to scale up to high-dimensional state spaces, either in continuous control (e.g. MuJoCo robotic environments) or in image-based tasks (e.g. video games). In this last part, we consider such environments, where RGL retains the same overall behavior, as its planning graph is defined over the goal state $G$ and not the state space. The key challenge in these environments is therefore to derive a good lower-level goal-reaching control policy $\pi(s, g)$, which is independent of the contribution of this paper and may require a large engineering effort, unrelated to RGL itself. In other words, the lower level policy in the Ant environment should learn to walk to any nearby position around its starting position, while RGL will enable it to navigate to distant positions in the maze.

Training a universal local goal-reaching policy in ant-maze environments, even in an obstacle-free playground, is already a challenging task. Appendix G expands on the pre-training procedure set in place to attain such a policy $\pi(s, g)$. HAC is the reference method for ant-maze environments but, as discussed in Appendix H, its efficiency appears brittle and despite our best efforts and the use of the original HAC implementation, it could not solve any of the ant-maze tasks. Similarly, the more recent (and comparable to RGL in spirit) DSG method reaches an extremely low accuracy, as reported in Figure 7 and discussed in Appendix I, where RGL eventually reaches very high levels of accuracy.

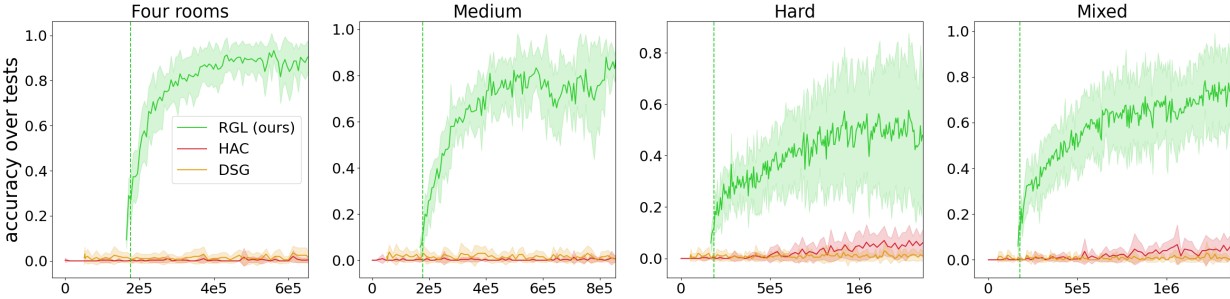

Figure 7: RGL accuracy in ant-maze tasks, as a function of training time steps. See Appendix H and I for a detailed discussion on HAC and DSG's low score.

|  | Four rooms | Medium | Hard | Mixed |
|---|---|---|---|---|
| RGL (ours) | **0.91(0.06)** | **0.86(0.06)** | **0.48(0.24)** | **0.73(0.16)** |
| HAC | 0.0(0.01) | 0.01(0.01) | 0.06(0.05) | 0.06(0.05) |
| DSG (Sparse rwd.) | 0.02(0.03) | 0.01(0.02) | 0.01(0.02) | 0.0(0.0) |

Table 4: Final average score for each method (standard deviation in parenthesis). Take-away: RGL statistically outperforms all baselines.

In comparison to the ant-maze environments used in Bagaria et al. (2021), the maps used here are larger. In turn, this makes reaching specific areas (goals) of the maze more difficult to reach. Besides, even when taking into account the pre-training time of RGL's lower-level policy, Figure 7 (and the original DSG paper) shows that DSG requires many more learning steps to correctly learn the individual options corresponding to the vertices in its planning graph. In contrast, RGL leverages the possibility to re-use the same TILO policy throughout the state space, and quickly grows the planning graph while leaving the lower-level policy unchanged after the first $20,000$ time steps. Additionally, DSG relies quite strongly on dense reward models for efficient training of options. The present environments avoid the (dense) reward shaping mechanism used in Bagaria et al. (2021)'s experiments, to avoid pre-training biases, which strongly penalizes DSG. Appendix I test DSG in various Ant-Maze settings to analyse the reasons of DSG under-performances in the current context.

Finally, ant-maze tasks, on top of being highly challenging for the baseline agent, also violate the assumption of Section 3.1 that any two states within $K_g$ are reachable from each other for a negligible cost. For instance, some ant orientations, velocities and leg configurations are rather complex to reach from others. Consequently, an edge between $g$ and $g'$ in RGL's graph only represents reachability of $g'$ from a subset of states in $K_g$, which can lead to plan failure (extended discussion in Appendix B). Despite this, RGL manages to achieve large values of accuracy almost as high as those obtained on point-maze tasks on the "four-rooms", "medium" and "mixed" mazes. The most challenging setting remains the "hard" maze, which requires fine motor skills to efficiently navigate through narrow corridors and requires turning around many corners to navigate to far goals.

**High-dimensional goal spaces.** The question of scaling up RGL to large goal spaces arises naturally as a follow-up to the previous paragraphs. For instance, when extending RGL from navigation to manipulation tasks, goal space dimension might increase and RGL's graph growth will suffer from the curse of dimensionality. This issue arises for all methods which design a graph in goal space; for instance, SGM or SoRB will

face the same issue as their state and goal spaces are identical. RGL's (slight) advantage here is precisely that the $\mathcal{P}$ projection enables building the graph in a lower-dimensional space than in the full state space, provided one could provide a suitable mapping for $\mathcal{P}$. We reserve the automated discovery of $\mathcal{P}$ for future work as it is a distinct challenge in itself.

## 7 Conclusion

Efficient coupling of planning and learning in complex MDPs with temporally extended goals is an active field of research. In this work we defend the idea that efficient mechanisms rely on two implicit hypotheses: planning agents should plan in the *goal space* and learned policies are often *re-usable* throughout the state space. We propose a formal framework accounting for these two notions, defining goals as state abstractions, and casting re-usability as translation invariance. This permits deriving an algorithm which performs planning over a graph of goal waypoints, reachable by a lower level goal-reaching policy. This agent is named RGL (*reachability graph learning*) after its training mechanism, which incrementally explores its environment using a pre-trained lower level translation invariant goal-reaching policy, expands and prunes a graph encoding goal reachability. This approach can be seen either as a more grounded version of STC (Ruan et al., 2022), or a generalization of SoRB (Eysenbach et al., 2019) or SGM (Emmons et al., 2020) to a hierarchical setting with translation invariance. Empirical evaluation confirms the relevance of RGL agents and their key features.

This contribution also forms a basis for future research directions. As is, RGL agents build a somewhat uniformly dense graph, as illustrated in Figure 2h. This might not be necessary and further sparsity can be achieved in the open rooms of the "four-rooms" or the "mixed" environments. Similarly, weight learning in the graph is currently rather naive and could better exploit interaction data during exploration, in particular in stochastic environments. The influence of playground environments on the final policies is also an open question, which connects to lifelong RL and curriculum learning: can one design good playgrounds for pre-training, the same way we let kids learn generic skills from simple (playground) tasks before we encourage them to compose these solutions together?

Finally, RGL requires an $\epsilon$-TILO policy for agent control. Appendix D and E propose a discussion on whether such policies are easy to obtain in the general case, beyond navigation tasks. We conjecture such policies also exist in more complex contexts, like vision-based navigation (PO)MDPs, such as those proposed by Kempka et al. (2016) or for real-life robots, since humans seem to exploit such invariances in daily life. Formalizing how these policies can be discovered and how their definition affects the properties derived in the present work is an exciting avenue for research.

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

## A  Hyperparameters and computational setup

Tables 5 to 12 summarize the hyperparameters used when training the different algorithms. The actor network used for the lower level goal-reaching policy takes a state and a goal as input (the dimension varies depending on the task) and processes them through a 2 hidden layer MLP. The output layer depends on the algorithm. Training follows the procedure of DQN and HER with discount factor $\gamma$ and exponential smoothing on the target network (factor $\tau$), and an Adam (Kingma & Ba, 2015) optimizer with default parameters. These parameters are the same for pretraining lower level policies for all algorithms. All RGL agents share the same $T_e$, $\eta_{node}$, $\eta_{edge}$, and $\eta_{reach}$ when applicable (for instance, PO-RGL uses $\eta_{edge}$ but not $\eta_{node}$ since it does not create new nodes). TC-RGL uses specific values $\eta_{node}$ and $\eta_{edge}$ for thresholds on node and edge creation, since it uses STC's temporal consistency network to measure node distance instead of our $d^\pi$ pseudo-metric; the scale of this network's output is unrelated to that of $d^\pi$ (hence the different values of $\eta_{node}$ and $\eta_{edge}$).

The results on grid-maze and point-maze were run on a desktop machine (Intel i9, 10th generation processor, 64GB RAM) with no GPU usage. The results on ant-maze were obtained with single node computations. Each of these nodes was composed of 2 12-core Skylake Intel(R) Xeon(R) Gold 6126 2.6 GHz CPUs with 96 Go of RAM (no GPU hardware).

Our code is available at [Anonymous URL].

## B  Complete results on reachability graphs

Figures 8, 9 and 10 present an extended version of the reachability graphs of figure 2 for all mazes in, respectively, grid-maze, point-maze and ant-maze environments. Blue dots in some figures correspond to the current selected goal at the time the graph was printed and should be discarded.

PO-RGL was created purely for didactic reasons in order to illustrate the pruning process independently of the incremental graph growth. Besides this illustration itself, these figures underline two features. First, the fact that RGL creates the graph incrementally makes it much sparser and avoids clusters of really close, redundant nodes. In turn, this sparse graph is much easier to prune than that of PO-RGL. Secondly, in environments with a fixed initial state (grid-maze, ant-maze), some edges never participate in the shortest path to any goal and hence are never pruned. Even if the sparse growth of the RGL graph limits this phenomenon, some impassable edges remain; e.g. some edges at the far right of figure 8p. Randomly resetting the starting state at each episode permits a more complete and easier exploration of all shortest paths, and hence results in a slightly more accurate pruning; e.g. the unpruned edges in grid-maze are better pruned on figure 9p.

Figure 10 (ant-maze environments) deserves a few additional comments. On this graph, to ease the readability and account for directed edges, whenever a directed edge exists between $v$ and $w$, we plot the edge's segment closest to $v$ in green. Orange then means the reverse edge has not been created. Red means the edge has been pruned. Some pruned edges appear in areas which seem passable. To explain this phenomenon, one needs to recall that the state space is 29-dimensional and a waypoint in the goal space (a geographical position of the ant's center of mass) can stand for a wide variety of configurations, as discussed in Section 4. For any two nodes $g$ and $g'$, it is possible that $g'$ was reachable from $\bar{\mathscr{P}}(g)$ but is not reachable from some other states in $K_g$, since ant-maze environments violate the hypothesis that all states in $K_g$ are reachable from each other for a negligible cost given the pre-trained policy. This leads to some edges being legitimately pruned while a "naive eye" laid on the reachability graph might conclude there was a mistake.

Finally, the graphs grown by RGL in ant-maze environments feature very few edges crossing walls. This is a side effect of the default values of $\eta_{node}$ and $\eta_{edge}$ (kept the same throughout all environments and mazes), and the fact that the ant's geometry prevents its center of mass to get close to the wall. This sometimes happens nonetheless when the ant randomly "tries" to climb over the wall (and systematically fails), which also places a few nodes that appear to be inside the walls.

Table 5: DQN hyperparameters. DQN+HER is used in grid-maze tasks to compute goal reaching policies.

| DQN | |
|---|---|
| model hidden layers | 64, ReLU, 64, ReLU |
| optimiser | Adam(lr=1e-3, betas=(0.9, 0.999), eps=1e-08, weight_decay=0) |
| replay buffer size | 1e5 |
| batch size | 100 |
| discount factor $\gamma$ | 0.95 |
| exponential smoothing factor $\tau$ | 1e-3 |

Table 6: SAC hyper-parameters. SAC+HER serves as a control policy for RGL, PO-RGL, and TC-RGL, as well as a baseline, in the "point-maze" and "ant-maze" environments.

| SAC | | |
|---|---|---|
| | Point-Maze | Ant-Maze |
| critic hidden layers | 250, Relu, 150, Relu | |
| actor layers | | |
| optimiser | Adam(lr=5e-4, betas=(0.9, 0.999), eps=1e-08, weight_decay=0) | |
| replay buffer size | 1e5 | 1e6 |
| batch size | 100 | 500 |
| $\gamma$ | 0.99 | 0.99 |
| $\tau$ | 5e-3 | 5e-3 |
| critic alpha | 0.6 | 0.6 |
| actor alpha | 0.05 | 0.1 |

Table 7: C51 hyper-parameters, which serves as a control policy for SGM in the "grid-maze" environment.

| C51 | |
|---|---|
| output distribution size | 20 |
| models layers | 64, ReLU, 64, ReLU |
| optimiser | Adam(lr=1e-3, betas=(0.9, 0.999), eps=1e-08, weight_decay=0) |
| replay buffer size | 1e5 |
| batch size | 100 |
| $\gamma$ | 0.95 |
| $\tau$ | 1e-3 |

Table 8: Distributional DDPG hyper-parameters, which serves as a control policy for SGM in the "point-maze" environment.

| Distributional DDPG | |
|---|---|
| output distribution size | 20 |
| models layers | 64, ReLU, 64, ReLU |
| optimiser | Adam(lr=1e-4, betas=(0.9, 0.999), eps=1e-08, weight_decay=0) |
| replay buffer size | 1e6 |
| batch size | 64 |
| $\gamma$ | 0.99 |
| $\tau$ | 0.05 |

Table 9: RGL hyper-parameters.

| RGL | | | |
|---|---|---|---|
| | Grid-maze | Point-maze | Ant-Maze |
| $\eta_{edges}$ | 0.2 | 0.045 | 0.3 |
| $\eta_{nodes}$ | 0.1 | 0.017 | 0.1 |
| reachability threshold of the nodes | 1 | 0.8 | 0.7 |
| max time-steps to reach next node | 50 | 50 | 150 |
| Exploration goal range | 2 | 4 | 6 |
| interactions per exploration | 90 | 90 | 150 |

Table 10: PO-RGL hyper-parameters.

| PO-RGL | | |
|---|---|---|
| | Grid-maze | Point-maze |
| $\eta_{edges}$ | 0.2 | 0.03 |
| reachability threshold of the nodes | 1 | 0.8 |
| max time-steps to reach next node | 50 | |
| nb nodes | four rooms: 400 medium: 600 hard: 600 mixed: 900 | four rooms: 400 medium: 500 hard: 700 mixed: 900 |

Table 11: TC-RGL hyper-parameters. Hyper-parameters that are not reported here are the same that the ones in Table 9 for RGL. "targeted edge length" is the minimum number of interactions that must separate two states of the same trajectory, so that they can form a positive pair (distant states) in the TC-network training data.

| TC-RGL | | | |
|---|---|---|---|
| | | Grid-maze | Point-maze |
| $\eta_{edges}$ | | 0.4 | 0.1 |
| $\eta_{nodes}$ | | 0.2 | 0.03 |
| TC-Network | layers | 125, ReLU, 100, ReLU, 1, Sigmoid | |
| | batch size | 250 | |
| | buffer max size | 1e9 | |
| | optimizer | Adam(lr=1e-3, betas=(0.9, 0.999), eps=1e-08, weight_decay=0) | |
| | targeted edge length | 20 | |

Table 12: SGM hyper-parameters.

| SGM | | |
|---|---|---|
| | Grid-maze | Point-maze |
| node pruning threshold | four rooms: 2
medium: 3
hard: 3
mixed: 2 | four rooms: 3
medium: 3
hard: 3
mixed: 3 |
| max edges length | four rooms: 5
medium: 6
hard: 6
mixed: 5 | four rooms: 7
medium: 7
hard: 7
mixed: 7 |
| nb initial nodes | four rooms: 1400
medium: 1400
hard: 1800
mixed: 1600 | |
| reachability threshold | 1 | |
| max interactions per sub task | 20 | |

## C  Complete results on graph sizes

Figures 12 and 15 complement Figures 4 and 5 by reporting the evolution of nodes/edges count across interaction time steps in point-maze environments. Note that graphs on point-maze environments required a log-scale on the $y$-axis for readability since TC-RGL spanned an order of magnitude more nodes than RGL (and two to three orders of magnitude more edges).

## D  Are $\epsilon$-TILO policies common?

In the present work, an important assumption is the existence of an $\epsilon$-TILO policy. Thus it seems important to discuss how restrictive this assumption is, and how commonly such policies might occur.

In position-based navigation tasks where $S = G$, generalization by translation invariance seems intuitive and easily justified by the translation invariance of the MDP's transition model properties throughout the state space. In navigation tasks where the state space is the agent's full configuration, but with abstract goal spaces (e.g. agent overall position), such as the ant-maze benchmarks, finding TILO policies is closely linked to defining the goal space, and hence the $P : S \to G$ projection. In this specific example, $P$ is defined by simply keeping some variables of $S$ and discarding the others. Here again, the TILO property is intuitive and translation invariance permits generalizing learned policies to unexplored parts of the state and goal spaces.

However, when it comes to state spaces with confounding variables, such as visual navigation tasks, then defining $P$ for abstract goal spaces might become more difficult as it links the input image pixels to positions on the navigation map. In a way, $P$ encodes expert knowledge about what abstractions of the state define a useful goal space, as discussed for instance by Forestier et al. (2022). Such abstractions might be learned (Péré et al., 2018) but since they are a pre-requisite for training a goal-based policy, they are generally considered to be provided by some expert. Such a description of goals is sometimes accessible for a minimal cost (as in navigation tasks), but a perspective for future work implies learning relevant goal descriptors from data. One one can draw a parallel with recent work in expressing goals with natural language and exploiting (large) language models to embed the goal description. Note, however, that in the general case, even if the corresponding $P$ encoding is given, there is no guarantee that a TILO policy exists.

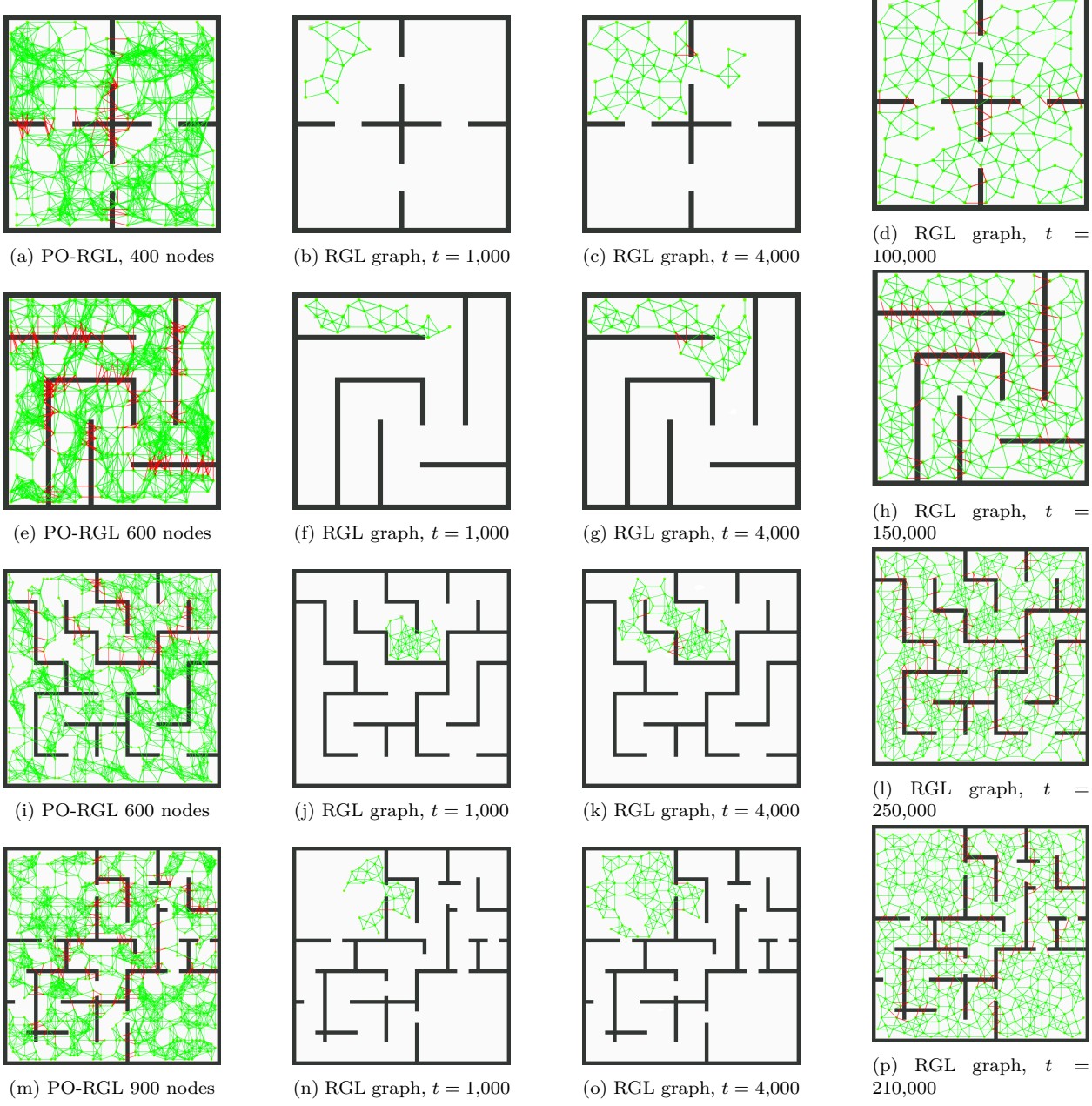

Figure 8: Reachability graphs, grid-mazes.

Besides the considerations above, we argue that the existence of TILO policies is intrinsically linked more to the nature of the task at hand than the definition of the goal space. Navigation is implicitly about finding a (potentially convoluted) path through a terrain with somewhat homogeneous properties. Hence, at least for this family of tasks, the existence of $\epsilon$-TILO policies is a plausible assumption.

## E From translation invariance to transformation invariance

Considering only translation invariance is somehow restrictive as one could wish to identify invariances to other transformations (e.g. deformations on images, rotations, etc.). In this section, we attempt to abstract the intuitions of translation invariance into a more general concept of *transformation* invariance.

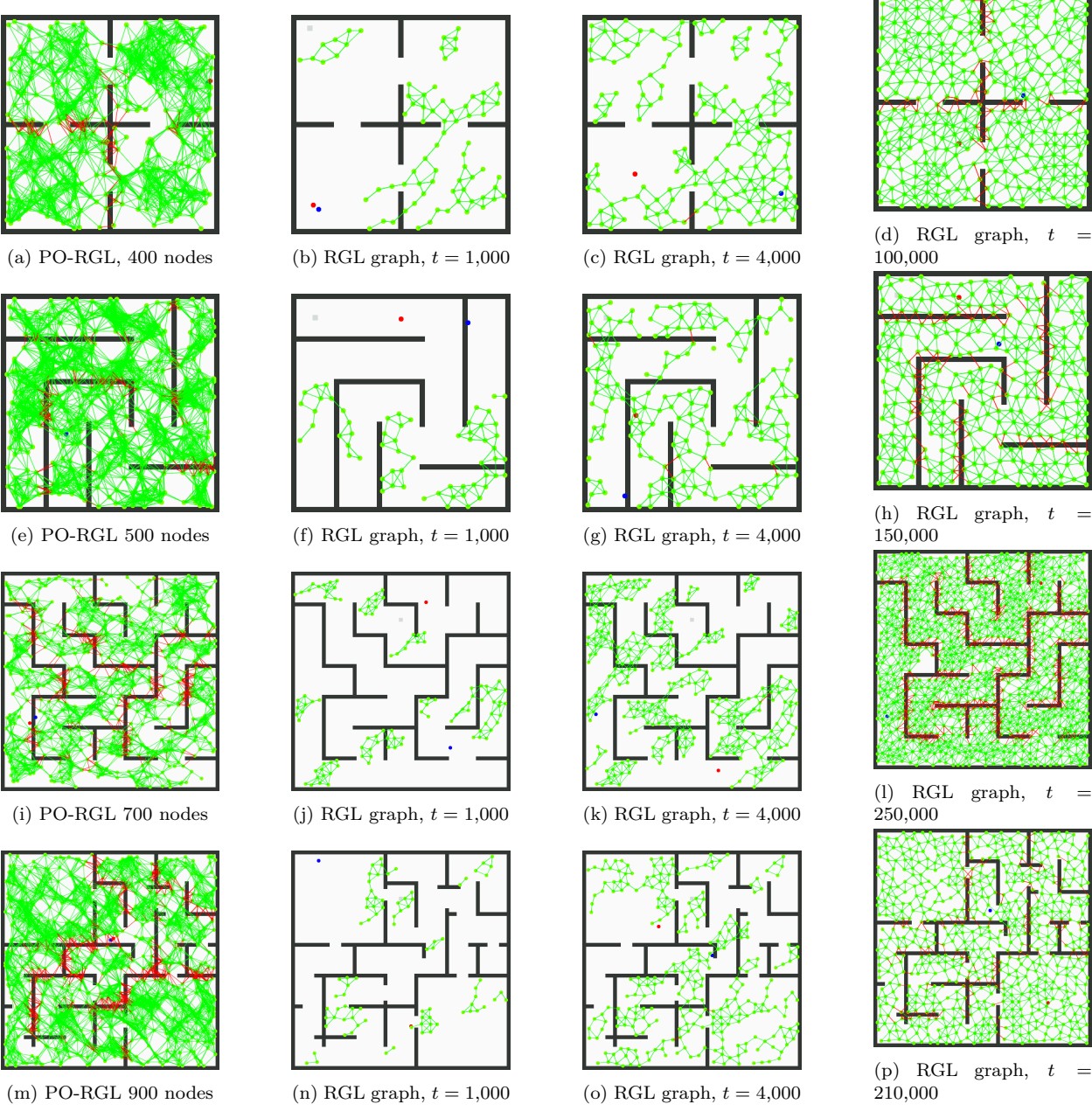

(a) PO-RGL, 400 nodes    (b) RGL graph, $t = 1,000$    (c) RGL graph, $t = 4,000$    (d) RGL graph, $t = 100,000$

(e) PO-RGL 500 nodes    (f) RGL graph, $t = 1,000$    (g) RGL graph, $t = 4,000$    (h) RGL graph, $t = 150,000$

(i) PO-RGL 700 nodes    (j) RGL graph, $t = 1,000$    (k) RGL graph, $t = 4,000$    (l) RGL graph, $t = 250,000$

(m) PO-RGL 900 nodes    (n) RGL graph, $t = 1,000$    (o) RGL graph, $t = 4,000$    (p) RGL graph, $t = 210,000$

Figure 9: Reachability graphs, point-mazes.

Suppose $\tau(s, \delta)$ is a transformation of $s$, with a parameter $\delta$. In Section 3.2, the transformation was a translation of vector $\delta \in S$, which defined $\tau(s, \delta) = s + \delta$. But as another example, when $s$ stands for geographic coordinates, $\tau(s, \delta)$ can be a rotation of angle $\delta \in [-\pi, \pi]$ around a central point. For genericity, we shall write $\delta \in \Delta$.

Then, to define the invariance of the policy to this transformation, we need the analogous transformation in the goal space, which we write $\tilde{\tau}(g, \delta)$. If $S = G$ this transformation is obviously $\tau$ itself. For translation invariance in the general $S \neq G$ case of Section 3.2, we defined $\tilde{\tau}(g, \delta) = g + \mathscr{P}(\delta)$.

Then, equation 1 can be re-written as

$$\forall s \in S, \delta \in \Delta, \exists \rho \in \mathbb{R}, \text{ such that } \forall g \in \mathcal{B}(\mathscr{P}(s), \rho), \pi^*(s, g) = \pi^*(\tau(s, \delta), \tilde{\tau}(g, \delta)). \tag{2}$$

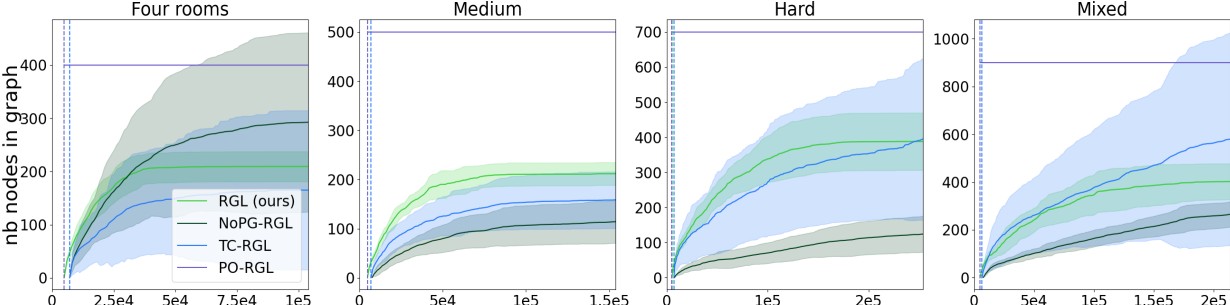

(a) RGL, $t = 6{,}000$

(b) RGL, $t = 24{,}000$

(c) RGL, $t = 500{,}000$

(d) RGL, $t = 6{,}000$

(e) RGL, $t = 24{,}000$

(f) RGL, $t = 700{,}000$

(g) RGL, $t = 6{,}000$

(h) RGL, $t = 24{,}000$

(i) RGL, $t = 1{,}200{,}000$

(j) RGL, $t = 6{,}000$

(k) RGL, $t = 24{,}000$

(l) RGL, $t = 1{,}200{,}000$

Figure 10: Reachability graphs, ant-mazes.

Figure 11: Number of graph nodes in grid-maze versus interaction steps. Shaded area is the $1\sigma$ confidence interval.

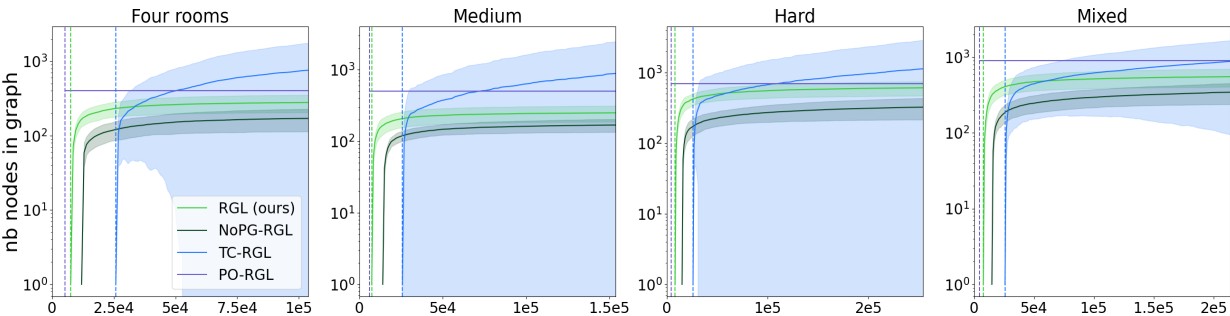

Figure 12: Number of graph nodes in point-maze versus interaction steps. Shaded area is the $1\sigma$ confidence interval.

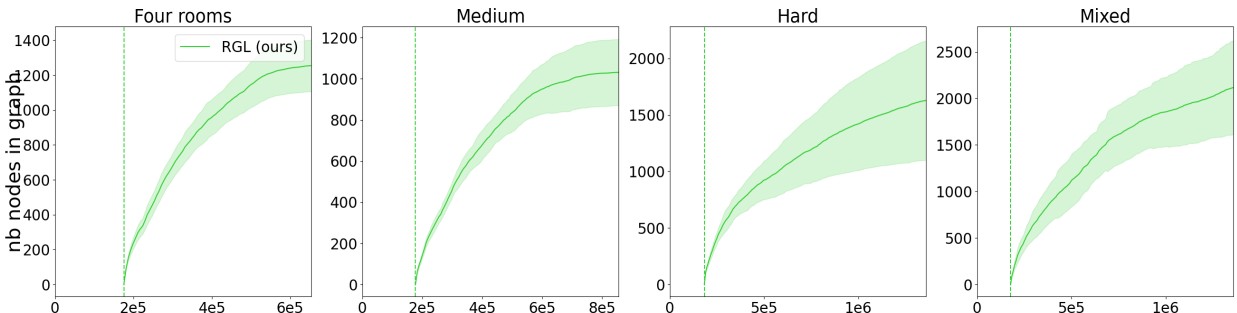

Figure 13: Number of graph nodes in ant-maze versus interaction steps. Shaded area is the $1\sigma$ confidence interval.

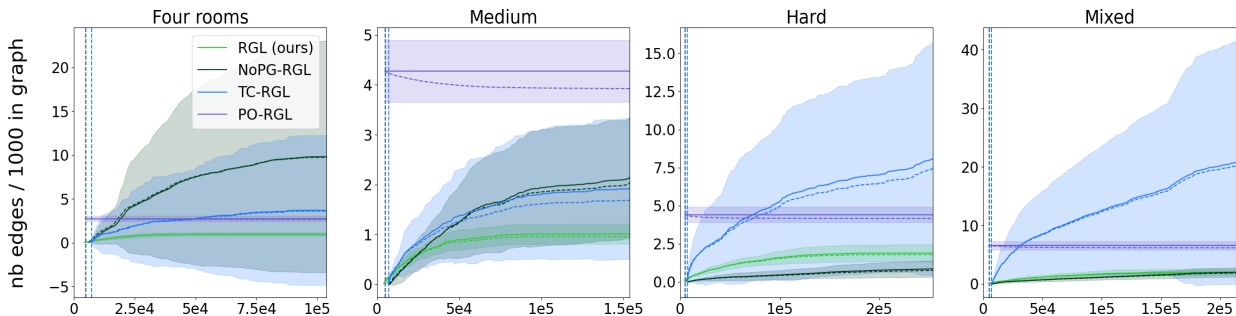

Figure 14: Number of graph edges in grid-maze versus interaction steps. Shaded area is the $\sigma$ confidence interval.

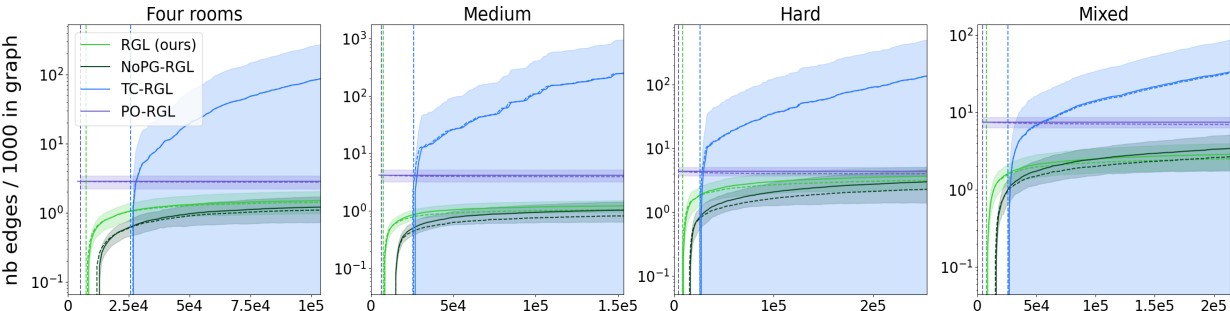

Figure 15: Number of graph edges in point-maze versus interaction steps. Shaded area is the $\sigma$ confidence interval.

An optimal policy respecting equation 2 can now be called $\tau$-invariant locally optimal ($\tau$ILO). Note that translation invariance was an intuitive and straightforward application of this definition, which holds in

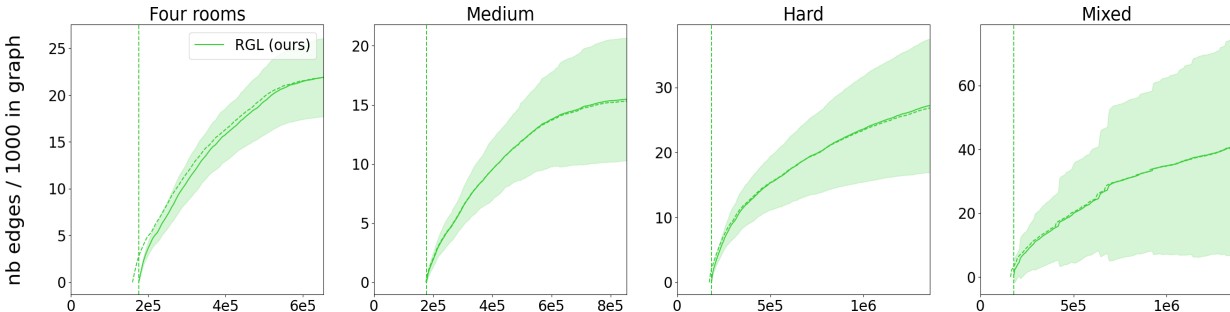

Figure 16: Number of graph edges in ant-maze versus interaction steps. Shaded area is the $\sigma$ confidence interval.

navigation tasks. Depending on the task at hand and the transformation $\tau$ considered, it might be very difficult to prove the existence of $\tau$ILO policies (Angelotti et al., 2022).

A special case of $\tau$ILO policies appears in visual navigation tasks, as those presented in the work of Eysenbach et al. (2019) or Emmons et al. (2020). Appendix J further discusses this specific case and the technicalities of training visual goal-reaching policies; in the present paragraph, we restrict ourselves to discussing why the corresponding policies are $\tau$ILO. In such tasks, the policy's input is a first-person view of a robot's surroundings, the goal space is the state space, the action space are relative movements, and previous works discard the fact that the stochastic process defined by the visual observations is likely not an MDP (rather a POMDP). Now, let us consider two different states $s$ and $s'$ of the robot in the maze, with the corresponding image observations $o(s)$ and $o(s')$. Suppose also that the relative positions of the walls around $s$ and $s'$ are the same, that is for instance, if the agent was facing a wall from a certain distance and with a certain angle in $s$, it is also facing a wall with the same distance and angle in $s'$. Then $o(s)$ and $o(s')$ differ only through graphical features like texture or lighting for example. If the policy (or value function) has been properly trained to detect the wall in the picture, regardless of its texture or lighting, then it will be insensitive to the texture or lighting change between $o(s)$ and $o(s')$, and hence will be able to generalize a goal-reaching policy trained in $s$, to $s'$. Formally, let $\tau$ be the transformation that turns $s$ into $s'$, then a locally optimal policy $\pi^*(s, g)$ will output the same action in $(\tau(s), \tau(g))$. These policies are $\tau$ILO by construction (because they rely on first-person views and because their neural network embeddings are supposedly trained to be insensitive to irrelevant image features) and respect equation 2.

## F   Influence of graph density hyperparameters.

The thresholds $\eta_{node}$ and $\eta_{edge}$ on node and edge creation condition how coarse the graph is in the goal space. Consequently, they impact the density of the graph, hence the ability to accurately represent transition dynamics. As such, they encode a notion of minimal required granularity to efficiently generate efficient goal-reaching plans in the goal space. Despite RGL's ability to build sparse representative graphs, a poor choice of $\eta_{node}$ and $\eta_{edge}$ parameters can be detrimental to RGL's goal reaching accuracy. Figure 17 reports how sensitive PO-RGL and RGL agents are to these parameters, in the "medium" grid-maze. Figure 17a illustrates how increasing the values of $\eta_{node}$ to 0.2 (then 0.3) and $\eta_{edge}$ to 0.4 (then 0.5) results in a graph which does not enable reaching distant goals anymore. A similar effect happens for PO-RGL, as choices for $N_{init}$ will have a direct impact on the accuracy of the algorithm. Figure 17b reports how varying $N_{init}$ from 100 to 600 affects the goal reaching accuracy of PO-RGL. With only 100 nodes, the reachability graph of PO-RGL features subgoals which are very distant from each other and rarely reachable between each-other, resulting in a graph with almost no edges (Figure 17c). Hence, no goals besides those reachable by the lower-level policy can be reached. With 200 nodes (Figure 17d), the final goal reaching accuracy of PO-RGL improves to about 50% and keeps improving until $N_{init} = 400$ nodes. For $N_{init} = 500$ and 600, the number of edges to prune in the graph becomes so large that it slows the learning down, resulting in less reachable nodes after 100,000 interaction steps because the graph contains too many misleading edges which have not been pruned yet. Overall, this illustrates how the directed, exploration-driven node and edge creation of RGL yields graphs which are both much sparser and much more representative of reachability, than building

a graph over randomly sampled goals (either randomly sampled from a replay buffer as in SoRB, or randomly sampled from an oracle as PO-RGL).

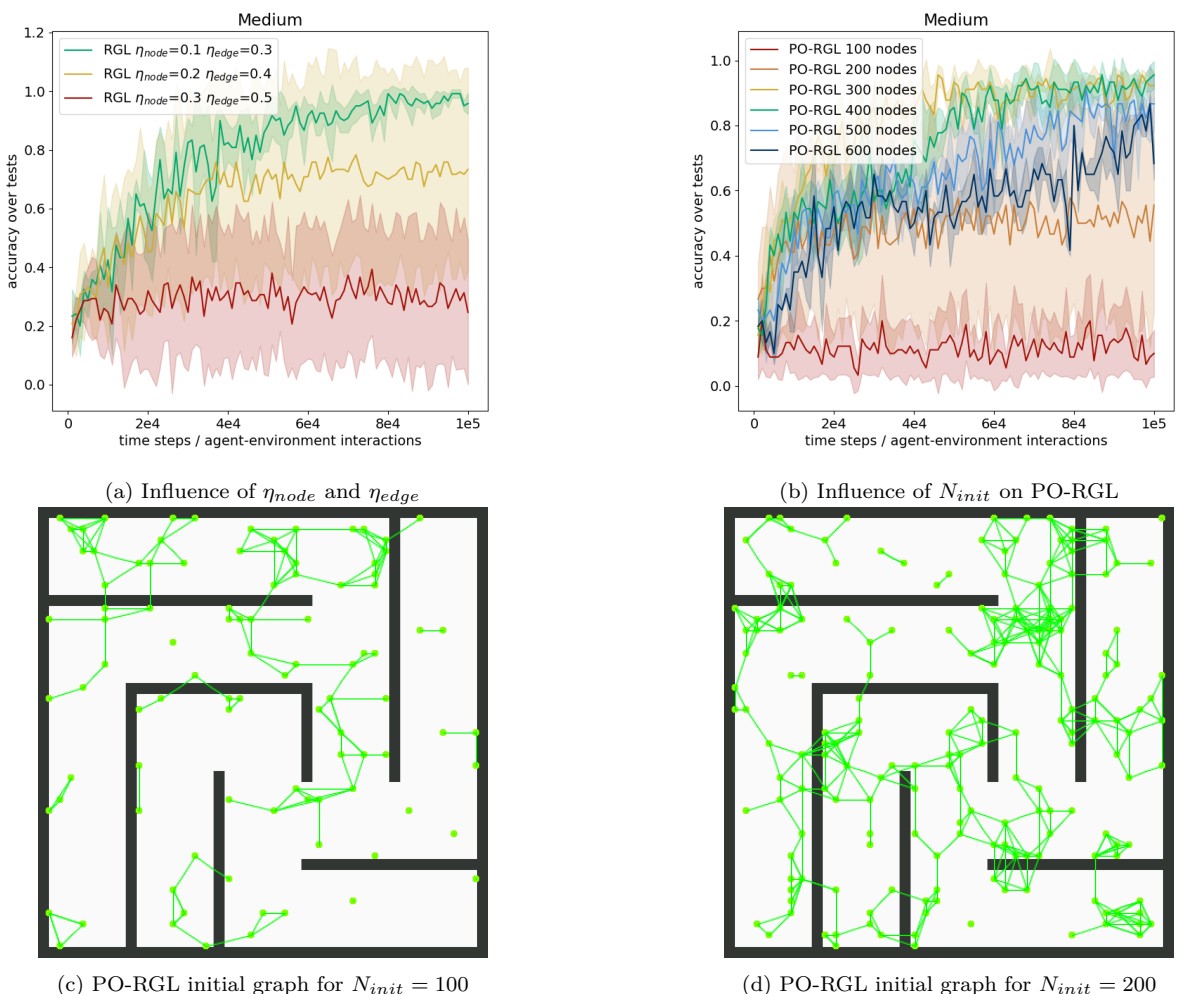

(a) Influence of $\eta_{node}$ and $\eta_{edge}$

(b) Influence of $N_{init}$ on PO-RGL

(c) PO-RGL initial graph for $N_{init} = 100$

(d) PO-RGL initial graph for $N_{init} = 200$

Figure 17: Hyperparameter influence on goal-reaching accuracy in the "medium" grid-maze after 100,000 interaction steps.

## G  Pre-training a goal-conditioned TILO policy in ant-maze.

Learning goal-based policies for ant-maze environments is challenging, even in the obstacle-free playground. To let SAC+HER converge efficiently, we build a process inspired from curriculum learning (Bengio et al., 2009). We sample goals uniformly in a disc around the agent, starting with a radius of 0. Every time the agent reaches a goal, we increment the radius of 0.1, and decrease it when it fails. If the radius reaches a value of 6, we stop incrementing and let the agent reach an accuracy close to 100% in this pre-training playground. Note that this value of 6 is much larger than that of $\eta_{node}$ and $\eta_{edge}$ (see Appendix A).

While navigating in the graph, following a sequence of sub-goals, the agent will change its direction many times in an episode. This may lead to more diversity in the states encountered while navigating the maze than those seen during the pre-training. To mitigate this aspect and improve state diversity during pre-training, every 5 episodes, instead of a full agent reset, we reset only the agent's position but retain the orientation, legs configuration and velocities from the last state of the previous episode.

## H    Hierarchical actor critic on various tasks

Ant-maze tasks have been tackled in previous work, notably in the important HAC (Levy et al., 2019) contribution, on similar tasks to those reported here, in particular the "four-rooms" maze. In order to provide a fair comparison with RGL, we used the reference implementation of HAC provided by the authors at https://github.com/andrew-j-levy/Hierarchical-Actor-Critc-HAC-/tree/master/ant_environments/ant_four_rooms_3_levels. This section discusses why this implementation (without modifications) fails on the tasks reported here.

**Goal and initial state sampling in HAC to promotes diversity.** In the original HAC contribution, during training, goals are sampled uniformly in the center of each room (red areas in figure 18), then initial states are sampled uniformly in the center of another room. This induces a variety of starting states and insures that starting states and goals are always at least one room away from each other. In turn, this promotes diversity in the replay buffers, which facilitate policy training. In the experiments reported in Section 4, we argued that this "reset anywhere" feature was a particularly favourable case for exploration.

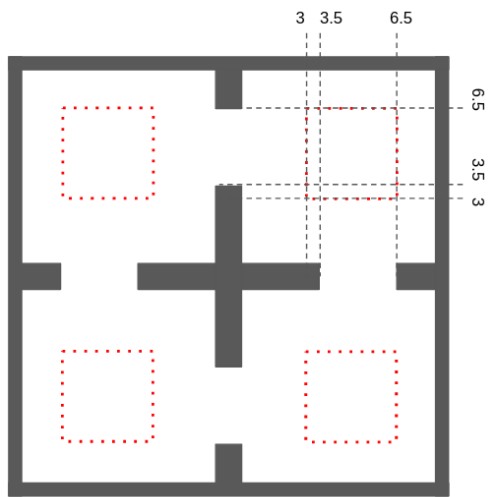

Figure 18: Illustration of goals and initial states sampling areas.

**Variations in mazes.** We also investigated whether the loss of efficiency of HAC could be attributed to the difference between the mazes presented here and those of the HAC paper. For this purpose, we tested HAC on three tasks and report results in Figure 19 (averaged over 10 trials).

1. The exact $17 \times 17$ "four-rooms" map used in the HAC paper, with the goal / initial state sampling strategy defined above (labelled *HAC sampling / small four-rooms* in Figure 19).

2. The same $17 \times 17$ maze map, but with uniformly sampled goals while keeping the starting state fixed (labelled *Uniform goals / small "four-rooms"* in Figure 19).

3. A larger $41 \times 41$ "four-rooms" map which is the one used in the RGL experiments of Section 4, with the HAC goal / initial state sampling strategy (labelled *HAC sampling / large "four-rooms"* in Figure 19). This map features slightly narrower passages between each room (proportionally to the size of the room). Actions remain the same: the ant is not scaled up. Goals are sampled uniformly. HAC's states and goals are scaled to the size of the map.

4. The same $41 \times 41$ "four-rooms" map with a fixed starting state and uniform goal sampling (labelled *Uniform goals / large "four-rooms"* in Figure 19).

The evaluation accuracy of each agent reported in Figure 19 follows the agent's goal / initial state sampling procedure than during training. Specifically, agents that were trained with the HAC sampling strategy are

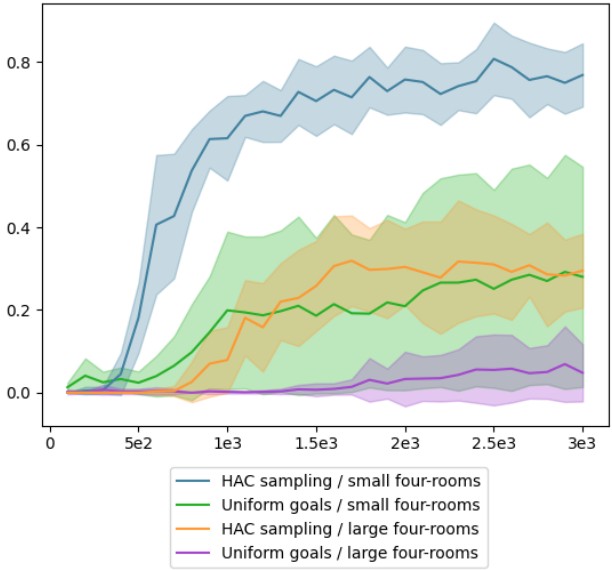

Figure 19: HAC average accuracy on variations of the "four-rooms" ant-maze task, versus number of episodes (episode length is capped at 700 time steps but can be smaller if the goal is reached before).

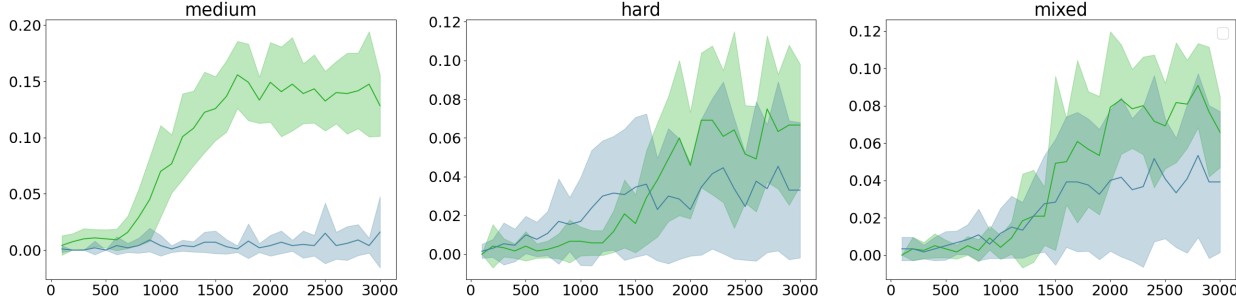

Figure 20: HAC average accuracy on the "medium", "hard", and "mixed" mazes for the ant-maze task, versus number of episodes (episode length is capped at 700 time steps but can be smaller if the goal is reached before). Green curve: uniform initial state sampling. Blue curve: fixed initial state.

evaluated by the proportion of reached goals when goals and initial states are drawn according to HAC's sampling strategy. Similarly, agents that were trained with a fixed starting state are evaluated on the same setting. Consequently, the only fair comparison with the RGL results of Section 4 is when the starting state is fixed and the goals are sampled uniformly. Recall that RGL reaches an accuracy of 89% on the large "four-rooms" environment with fixed initial state and uniform goal sampling (Figure 7).

It appears that the goal / initial state sampling strategy is a crucial feature of HAC in ant-maze. Removing this feature, and sampling goals uniformly, reduces the accuracy of HAC's optimized policy from 76% to 28% in the small "four-rooms" environment, and from 29% to 5% in the large one.

It also appears HAC is rather sensitive to the scale of the map (despite appropriate state scaling in the inputs of the neural networks): even with the HAC initial state / goal sampling strategy, the accuracy of the optimized policy does not exceed 28% (versus the 89% of RGL). More steps are required to cross a room between passages and we hypothesize HAC suffers from this difficulty to span long trajectories between goals and hence struggles to reach good accuracy in larger mazes.

Note that the HAC sampling strategy is tailor-made for the "four-rooms" maze and is undefined for other mazes, so the comparison above cannot be reproduced for the "medium", "hard", and "mixed" mazes. Instead (and this goes beyond what was proposed by the HAC authors), in an attempt to have a comparison baseline,

we replaced this HAC sampling strategy by uniform sampling of both the initial state and the goal in these three mazes. We also evaluated the fixed initial state / uniform goal sampling setting. Results (Figure 20) on other maps are similar to those of Figure 19: HAC reaches very small accuracy levels compared to RGL, even with the diversity of initial states and goals induced by uniform sampling. For this reason, we chose not to include these results in Section 4.

## I   Deep Skill Graphs on various tasks

This section discusses the similarities and differences between DSG and RGL. While there are similarities in their design principles, the two approaches remain quite different.

The main difference lies in the fact that DSG trains a different option for every vertex in the graph, following DSC's procedure (Bagaria & Konidaris, 2019). In contrast, RGL does not interleave lower-level policy training (which is a local goal reaching policy instead of a set of options) with graph expansion. This has the drawback (for RGL) of preventing local adaptation of behaviors, but the advantage of sample efficiency and a full re-use of pre-trained policies (which is the key interest of the TILO abstraction).

Beyond this key difference, initiation and effect sets of DSG play the same role as the neighborhoods defined by $\eta_{node}$ and $\eta_{edge}$ in RGL.

Aside from these elements, one should note that RGL generalizes from the $S = G$ context to contexts where $S \neq G$.

Also, and quite importantly, DSG uses the Euclidean distance between states as a reachability heuristic in graph construction, while RGL uses the value function $V(s, g)$ – which is somehow more principled. This distance is also used to define a dense reward model when training options.

This last point is particularly important in explaining the difference in empirical scores between DSG and RGL (see Figure 7). As indicated by Bagaria et al. (2021, Section 4), DSG makes use of dense rewards and this is where it really shines (Section 5 of the DSG paper actually discusses shortly the need to extend to sparse rewards). In the experiments reported in the present paper, the rewards are sparse, as we are in a purely goal-reaching policy search, where we did not introduce any reward shaping.

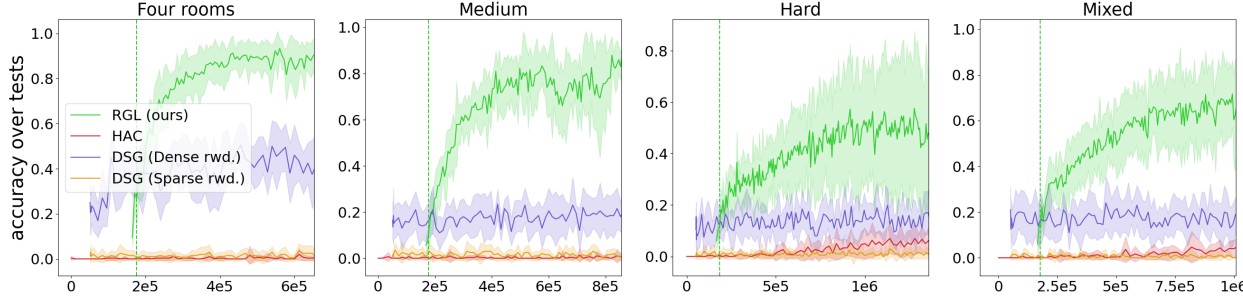

Figure 21: Accuracy of various methods in the Ant-Maze, using the same maps used in figure 7. Every methods are trained using sparse rewards, except the experiments labelled "DSG (Dense rwd.)

Figure 21 reports the difference in performance of DSG in the 4-rooms environment, both with dense and sparse rewards. In the case of sparse rewards, using DSC's procedure to re-train local options for every graph vertex might quickly become inefficient and, in contrast, the TILO policies of RGL appear as a better option in this context. In comparison, performing the same experiment with sparse rewards leads to an accuracy gap, and shows the reliance of this method for dense reward. However, the accuracy observed for DSG with dense reward is still far from the accuracy presented in the original paper. According to the authors, this method can fail to reach some goals while using dense rewards, especially when the goal to reach is at the other side of a wall.

However, the learned value function is less influenced by this reward the further away the goal is from the agent. Such problems are more likely to occur when both the goal and the agent are close to the wall. In

our experiments, the goals are sampled uniformly in the reachable space, and walls are thinner (in mujoco units, our mazes have a thickness of 1, while the ones used in DSG paper have a thickness of 4). We consider that such walls thickness could lead to a higher chance of seeing the DSG agent failing while trying to reach a goal by trying to cross a wall. DSG suffers from the same issue, but it can be easily corrected by adjusting the graph density hyper-parameter $\eta_{node}$, since RGL agent will only try to reach a goal from the closest node. Figure 22 shows the accuracy of DSG in a U-Maze with thick walls. The only difference between this experiment and the one called "DSG (Dense rwd.)" in figure 21 is the walls positions and their thikness.

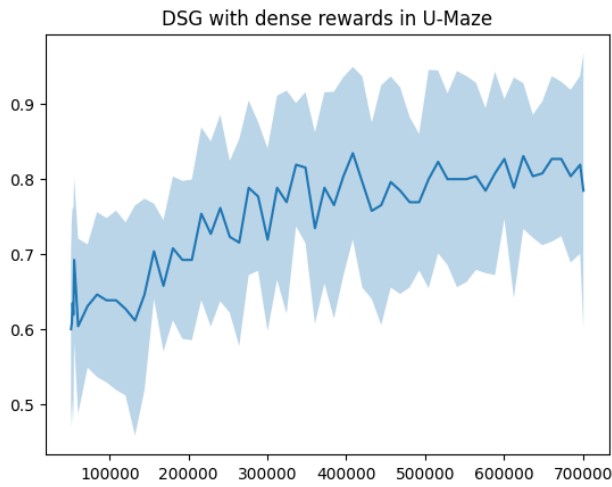

Figure 22: Accuracy of DSG with dense reward in a U-Maze with a wall thickness of 4

## J    Training in playgrounds and generalizing to unseen states

In Section 4 we claimed that "only RGL can, by design, exploit a goal-reaching policy which has not been trained on the actual full maze". This contrasts with, for instance, Section 5.5 of the SoRB paper (Eysenbach et al., 2019) where a visual policy is first trained to navigate in 100 houses from the SUNCG dataset from first-person images, then this policy's value function is transferred to new houses where it enables building a naviagtion graph. We argue the claim of Section 4 still holds and introduce nuance and justification below.

It is important to note that (i) this visual policy is very close to being TILO (see Appendix E) and, (ii) more importantly, is trained on data that covers reasonably well the set of images the agent might encounter. This second property makes the policy likely to generalize well to new houses in the first place. Consequently, although SoRB is indeed tested on new mazes, the combination of a first-person visual value function and the fact that training covers a representative set of images creates a favourable setting which implicitly captures translation invariance. We expand on this aspect in the following paragraphs.

SoRB and SGM are designed to build a graph on states from the pre-training replay buffer or on states that are close enough to those of the pre-training replay buffer, when such states can be drawn from an oracle (which is what happens in the experiment of Section 5.5 in the SoRB paper). This constraint is lifted with RGL thanks to the TILO property which nicely decouples local TILO skills learning from macro-action graph building. Specifically, for the transfer to new houses in the SoRB paper, it is important to examine how the value function is trained. In a nutshell, the authors of SoRB take $(s, g)$ pairs that are about 4 steps away in many different houses, and train a value function that indicates "how many consecutive actions does it take to transform this $s$ image into this $g$ image?". We conjecture that the combination of a "reset anywhere" possibility, plus training on 100 different houses, permits seeing a diversity of $(s, g)$ pairs that enables generalization to similar images in new houses (unfortunately the SUNCG dataset is not available anymore to verify this conjecture, due to legal issues). Then, SoRB's MAXDIST parameter is set to 3, which means SoRB creates an edge between two nodes whenever the value function estimates one needs at most 3 actions to connect the two corresponding states. Our point is that the diversity of $(s, g)$ pairs sampled

from the first 100 houses actually reasonably covers the set of possible images and that the pre-trained value function is a good estimator of "can I reach $g$ from $s$ in less than 3 actions?", even for images from different houses, that were not seen during training.

From this perspective, SoRB's value function in this precise case captures the notion of translation invariance without stating it explicitly. Once this is made clear, then what SoRB really does is sample states from an oracle (the reset function in the environment) and connects them using this value function, while SGM becomes almost equivalent to PO-RGL: it builds the same graph as SoRB and then prunes it. What enables SoRB and SGM to perform this graph building is precisely because the value function was trained on rich enough data to be able to evaluate distances between images sampled from new houses. Without this "implicitly-TILO" property, they wouldn't be able to build this graph reliably.

In conclusion, the exploitation of translation invariance in RGL can be interpreted as a principled generalization of the previous example: it permits pre-training the lower-level policy in a playground (such as the 100 first houses of the SUNCG dataset) and transferring the pre-trained value function to unseen states in another MDP, where the TILO property ensures to retain local optimality of policies. In turn, this decouples, in a principled way, learning lower-level skills (like walking in ant-maze, regardless of locally specific surrounding obstacles) from chaining skills and navigating.

