# OpenReview forum: "Learning State Reachability as a Graph in Translation Invariant Goal-based Reinforcement Learning Tasks"
_TMLR — Accepted by TMLR_

### Review · Reviewer_Pgxz · 2024-07-16

**Summary Of Contributions:**

This work proposes a method for goal-based RL in the context of environments which enjoy a translational invariance  (TILO) property (essentially, the assumption of the existence of ($\epsilon$-)optimal policies which are "smooth" with respect to the state/goal space). In particular, the proposal is to abstract the learning of low-level translation-invariant policies into a pre-training phase on an environment amenable to efficiently learning these low-level policies, followed by a planning on a reachability graph derived from the pre-trained policies. The key novelty in this work appears to be the introduction of the TILO property as well as an approximate and transformation versions thereof, which the authors use to develop their method. The method also combines several techniques existing in prior work and combines them together to achieve improved empirical performance on a set of maze tasks. The authors also perform several ablations of their method and compare to existing methods in their settings.

**Audience:**

Yes

**Claims And Evidence:**

Yes

**Requested Changes:**

Overall, I would vote to accept the paper assuming the critical questions below are resolved.

Critical Questions/Change Requests:
* Why is DSG achieving zero performance in Fig 7? Given that there is such an extreme gap in performance on the tasks evaluated in this paper, it would be good to see an evaluation of the presented methods on the specific tasks in the DSG paper or an explanation and analysis similar to Appendix H explaining the non-performance of DSG. At least, some more evidence is required for explaining the near-zero performance of DSG beyond an implementation error.
* It would be good to expand on the differences between the proposed method and DSG further. Since the methods are somewhat similar, it is surprising that DSG would perform so much worse.
* It would be good to lighten the claims in the introduction and emphasize the focus on the setting to which we know TILO properties apply. It is fine to highlight that the method could be more broadly applicable, but it should be emphasized that it is an adhoc procedure to determine a relevant "playground" (though I understand that one could always use the existing distribution of environments as a playground, it is still not clear in general how NoPG training generalizes beyond the settings studied in this paper as it is an empirical question).

Suggestions:
* The underperformance of NoPG-RGL compared to RGL in Fig.3 implies that in the main environment of the actual maze, it is harder to learn the movement policies useful for navigation as compared to the playground. Thus, is it fair to interpret this effect as an instance where completely decoupling the learning of a set of useful low-level skills adds significant information to the problem? Are there examples of cases where training on a “playground” like environment significantly hurts performance compared to training on the distribution of actual environments? Are there ways of detecting whether a “playground environment” will be helpful or harmful prior to running the full experiment?
* The discussion in Appendix I was quite interesting and helpful, and I wonder if the authors have any more comments on detecting the utility of and/or building potential playgrounds (are there simple tests that might enable one to determine whether a playground is useful? How about constructing playgrounds?)
* The paper would be greatly strengthened if more insight was given into how one could identify/develop useful playgrounds or at least efficiently test whether a playground is useful quickly, though I understand this may be outside the scope of the current paper.

Nits (see beginning of Section 3):
* I disagree with the statement: "continuous universal approximators, neural networks are intrinsically unsuited to approximate abrupt environment of skill changes, and discontinuous functions such as the value functions arising in some difficult RL environments" - it should be framed differently. The problem which arises doesn’t have to do with the fact that neural nets are continuous universal approximators, has to do w/how smooth a function it is, which can be controlled, and the difficulty of learning such functions as dependent on these smoothness parameters. As a simple counterexample, language is discrete and discontinuous yet we have LLMs.
* I disagree with the statement: "Because neural network optimization assumes that samples are obtained independently and identically from a stationary distribution, they are also unsuited to retain local information." This misattribution issue has nothing to do w/using neural networks or not, it has to do w/what learning problem you set up. Please rephrase or remove - using the wrong learning framework w/a neural net policy class doesn’t imply that the policy class is bad.
* "Neural networks are great at learning complex continuous functions such as navigation or movement primitives, or local goal-reaching policies." This statement also doesn’t make sense as-is and needs to be considerably clarified. I would discard all of these statements, the only point you need to make is that neural networks are a viable policy class for low-level policies and that it may be advantageous to add additional known structure to the problem setting (which can be learned) rather than attempting to learn a complete world model with no additional assumptions which may require high sample-complexity.
* "When requiring to achieve" → “When we would like to reach”
* "its ability to represent accurately an abstraction of the environment dynamics despite unbalanced samples and discontinuous properties" → accurately represent

**Strengths And Weaknesses:**

Strengths
* Definition of TILO is a simple and reasonable abstraction in many settings.
* Policy training pre-training method inspired by TILO (enforce translation property holds for resulting pre-trained policy by training policy as a function of $g - s$)
* Policy pre-training performed on a simple playground (though this is highly task-specific, the contribution would be significantly stronger if a more general approach to determining and evaluating playgrounds was proposed).
* A specific graph reachability approach is proposed, with dynamic growing and pruning phases. However, I should also note that the proposed method itself for graph reachability does not strike me as particularly novel though it does seem distinct from existing methods.
* Many experimental ablations demonstrating the strength of the proposed approach on several varying-complexity maze environments, including continuous and stochastic high-dimensional settings. A lot of the value in the paper is in these ablations and empirics.

Weaknesses
* It is hard to apply the TILO property in general, and requires significant work and insight to determine a playground for pre-training. Thus it is not clear how far beyond the existing settings studied the approach will generalize, though there are some gestures at applications to image-based settings in Appendix I.
* DSG, which is mentioned in the paper, seems quite similar in overall high-level concept. While Figure 7 suggests that the presented method drastically outperforms DSG, it is a bit puzzling since the DSG method is overall quite similar.
* Figure 17 suggests that hyperparameters need to be set quite carefully based on the environment to see a good performance, limiting direct applicability but still providing knobs to turn.
* There is an additional strong assumption that moving between states which correspond to the same goal is feasible and costless. This seems ok if there’s only really one way to reach a certain goal which has no influence on future strategies (e.g., acquiring some resource and state doesn’t matter for future steps), but it seems like an issue in harder settings. However, it is also nice to note that the ant-maze setting breaks this assumption (ant state configuration is independent of physical location in maze), but the proposed method still performs reasonably well.

---

> ### Author Response · Authors · 2024-08-21
> **Paper updated with additional experiments.**
>
> # On the difference(s) with DSG, both on algorithmic principles and empirical evaluation
>
> Although this work was completed independently of DSG, we agree there are similarities in the principles at stake, and have updated the paper to better highlight the differences.
> Specifically, to avoid overloading the main text, we have added a new appendix section about DSG and referred to it from the main text (Appendix I).
>
> In a nutshell, the main difference is that DSG trains a different option for every vertex in the graph, following DSC's procedure [Bagaria and Konidaris, 2020]. In contrast, RGL does not interleave lower-level policy training (which is a local goal reaching policy instead of a set of options) with graph expansion. This has the drawback (for RGL) of preventing local adaptation of behaviors, but the advantage of sample efficiency and a full re-use of pretraining (which is the key interest of the TILO abstraction).
>
> In turn, the cause of RLG's empirical domination of DSG lies in DSG's poor performance with sparse rewards, in conjunction with its interleaving of option training and graph expansion. The paper now features more discussion on these aspects and we hope this clarifies things (see Appendix I). Thank you for pointing this need for clarification.
>
> # On the importance and influence of playground environments
>
> We confirm that identifying or designing good playgrounds is an open question. Indeed, noPG-RGL illustrates that pre-trained policies from different playgrounds will induce different final scores for RGL, as they will have different local goal-reaching abilities. We see this more as a life-long training question than a weakness itself and have tried to make this more explicit in the introduction and in Section 3.2. The playground stands for previous MDPs seen by the agent, and the TILO property captures the idea that translation invariance is an abstraction that enables transfer between MDPs. From this perspective, designing good playgrounds connects to the problem of designing a curriculum of pre-training environments, which is beyond the scope of this study (even though we completely agree this is an important and exciting perspective). We have also modified the conclusion to better reflect this.

---

> > ### Comment · Reviewer_Pgxz · 2024-09-20
> > **Thanks for updates**
> >
> > Thanks for adding the updated section, it is helpful. I would still like the nits to be addressed however especially in Section 3, they are misleading as is - this should be an easy fix. My other concerns are resolved and I vote to accept.

---

> ### Author Response · Authors · 2024-09-30
> **Missing requested modifications included**
>
> We modified section 3 to include the missing modification. Thank you for your useful feedback.

---

### Review · Reviewer_JxoC · 2024-07-18

**Summary Of Contributions:**

The paper proposes a new method to learn control policies for reaching goals based on what the paper describes as reusable skills. To do so, the paper proposes using a graph structure that encodes abstract goals and enables planning in the goal space. The paper starts by introducing how reinforcement learning (RL) has shown promise in learning well-defined skills and proposes using the translational invariance of those skills to perform tasks in settings that should be easily translatable to situation that are practically not very different. The paper then defines its main contributions related to: 1. A generic framework linking goal space and states; 2. Providing a definition of reusability for goal-reaching policies; 3. A graph-based model training method. Section 2 summarizes related work, including goal reaching in RL, hierarchical RL, planning and reusable policies. Section 2 also details how the proposed method (RGL) is different from prior work.

Section 3 describes relevant definitions for the proposed method. The first definition relates to thinking of goals as state abstractions where one can define mapping functions between goals and a set of states in the environment. The definition of goals is then used to define the translation invariance of local optimal policies, which enables translating skills to reach practically similar goals. Next, the paper defines the graph topology that provides insight into reachable states from the current state and builds a graph that can be used for planning. Based on these three components, the paper defines its main algorithm: Reachability graph learning (RGL).

Section 4 describes the experimental setup, which focuses on different maze settings. The section defines the baselines that are compared against RGL and outlines a pre-training method that infuses the relevant skills into the goal-reaching policy. Section 5 outlines the main experimental results starting with visualizations of goal-reaching graphs across the different maze settings. The paper provides an analysis and details on ablation of the method in Figure 3, Figure 4 and Figure 5. Section 6 describes experiments on more challenging tasks using the ant environment in a maze comparing different ablations and baselines. Section 7 provides a conclusion and discussion.

**Audience:**

Yes

**Claims And Evidence:**

Yes

**Requested Changes:**

**Important Changes:**
* Please provide more descriptive figure captions that summarize the main settings and takeways for each. This will make it easier to understand the relevant context of the results.
* Please summarize the different ablation in a table or another way to easily find them. They way they are currently presented makes it difficult to track what is what.
* Please address the variance in the results plots. It appears that in some cases your method has distinction from the baselines, but I would recommened greater clarity on the ablation and where RGL provides clear benefits.

**Nice to Have:**
* A visual example of the reachability graph pruning and expansion (this can be in the paper or appendix).
* A discussion on applications of RGL - can be mainly in control problems but potentially also broader.

**Strengths And Weaknesses:**

**Strengths:**
* The paper provides useful definitions of relevant concepts (goal to state mapping, translation invariance) and detailed descriptions of the method.
* The related work section and context for prior work is quite detailed.
* The graph results in Figure 2 could be useful from an interpretability to understand what policies are capable of.

**Weaknesses:**
* Based on the results presented in Section 5 and Section 6, it is unclear if RGL provides benefits compared to other methods. In some cases the baselines are within the margin of error.
* The ablations often show unclear results about the most important parts of the method. The large variance shown in the learning curve applies to both the baselines and the ablations.
* The clarity could be improved for the experimental section. For example, the figures could be improved with more detailed captions and the paper could be shortened to fit into 12 pages to make the important takeaways more concise.

---

> ### Author Response · Authors · 2024-08-21
> **New materials and paper updated.**
>
> We thank the reviewer for these insightful comments which helped us improve the clarity of the paper.
>
> In the updated version, we have better separated the presentation of ablations (Section 4) to make them more apparent. We have also improved the figure captions to better reflect the take-away messages and added tables to make the final average scores and their standard deviation more readable and interpretable.
>
> We have also included in the supplementary materials, an archive of one video for each benchmark, representing the RGL graph expansion, to complement Figure 2. On these videos, we can see nodes, and edges in green, and pruned edges in red.
>
> We hope these edits address these useful remarks.

---

### Review · Reviewer_R9XL · 2024-07-22

**Summary Of Contributions:**

- This paper proposes a graph-based RL method that uses a goal-conditioned RL policy and state reachability to form a graph to solve RL problems.
- The paper formalizes the notion of translation invariance that enables the use of the goal graph as an abstraction of the original MDP.
- The proposed method can solve challenging long-horizon RL tasks.

**Audience:**

No

**Broader Impact Concerns:**

no broader concerns.

**Claims And Evidence:**

Yes

**Requested Changes:**

The author should at least include a discussion about the relative work mentioned above. Experiments or discussions about high-dimensional goal space are also required.

**Strengths And Weaknesses:**

The direction of combining the traditional graph-based search method and the RL is promising, and the paper demonstrates its effectiveness. I also appreciate the author’s definition of translation invariance, clarifying the conditions for which such a graph-based method can be used.

However, the paper should have included several important relative works in hierarchical imitation learning. For example, [1] also demonstrates an approach that builds a graph using a replay buffer and uses shortest path search to find sub-goals for the low-level RL policies. It also leverages the Q or value function as a quasi-metric between subgoals. Except for the tiny differences in how the graph is expanded ([1] simply sampling landmarks using farthest point sampling), most algorithms are quite similar.  Given the similarity between the two algorithms, this paper’s novelty is limited mainly.

[1] Huang, Zhiao, Fangchen Liu, and Hao Su. "Mapping state space using landmarks for universal goal reaching." Advances in Neural Information Processing Systems 32 (2019).

Another major limitation of the work is that only the navigation style environment is considered, where a low-dimensional goal space can be defined trivially. However, in most manipulation tasks, especially tasks that involve multiple objects, the goal space and state space become high-dimensional. Building such a graph becomes intractable.

---

> ### Author Response · Authors · 2024-08-21
> **Updated paper**
>
> We thank the reviewer for these remarks.
>
> We have included the suggested paper in the related work section. In fact, Huang et al approach has many points in commons with SoRB. The main differences are that they use a different node sampling strategy, and that they use, like RGL, a policy to reach sub-goals.
> However, this algorithm builds its graph on the control policy replay buffer, which limits it's exploration ability to the control policy one, like for SORB and SGM.
>
> We agree that RGL in high-dimensional goal spaces will suffer from the curse of dimensionality. We have added a paragraph at the end of Section 6 to discuss this point.

---

### Decision · Action_Editor_UQuy · 2024-10-22

**Recommendation:** Accept as is

**Comment:**

I recommend accepting this paper. The work introduces a well-formalized framework through the Translation Invariance (TILO) property and demonstrates its effectiveness through a novel graph-based RL method. The empirical results show promising performance on long-horizon tasks in maze navigation environments. Initial reviewer concerns about result variance and baseline comparisons were adequately addressed through additional experiments and clearer result presentations in the revisions. The authors have been responsive to reviewer feedback.

While the practical impact for real-world applications remains to be seen, and there are limitations in generalizability beyond navigation-type tasks, these uncertainties are well-acknowledged by the authors. The paper makes meaningful contributions to goal-conditioned RL and long-horizon planning, and could serve as a foundation for future research in this direction.

**Audience:**

Based on the reviews, this paper would be most valuable to researchers and practitioners working in reinforcement learning, particularly those focused on goal-conditioned RL and long-horizon planning problems. The work introduces theoretical concepts (like translation invariance) that would interest theoretical RL researchers, while also providing practical implementations that could benefit those developing autonomous navigation systems or working on similar spatial planning tasks. The clear formalization of the concept of "reusable skills" through the TILO property makes it especially relevant for those working on skill transfer and hierarchical RL. However, as noted in the reviews, the current implementation primarily demonstrates success in navigation-style environments, so it would be most immediately useful to those working in this domain, though the theoretical framework could potentially be extended to other areas.

**Claims And Evidence:**

The following are claims made in the submission. Overall, the paper presents an interesting theoretical algorithm with practical implementation, and the main concerns by the reviewers are on limited experiment tasks and noises in results.

Development of a graph-based RL method using goal-conditioned RL policy and state reachability
- Evidence: Implementation and experimental results in maze environments
- Support: Strong, demonstrated through multiple environments and comparisons

Translation Invariance (TILO) property formalization
- Evidence: Mathematical definitions and proofs
- Support: Strong, reviewers accepted the theoretical foundations

Method can solve challenging long-horizon RL tasks
- Evidence: Results on ant-maze environment and complex mazes
- Support: Mixed - showed good performance but with high variance in some cases

Proposed method outperforms baselines, particularly DSG
- Evidence: Comparative experimental results, especially Figure 7
- Support: Initially questioned but later strengthened through additional experiments/explanations